# A method to encapsulate model structural uncertainty in ensemble projections of future climate: EPIC v1.0

Jared Lewis[1], Greg E. Bodeker[1], Stefanie Kremser[1], and Andrew Tait[2]

[1]Bodeker Scientific, 42 Russell Street, Alexandra, 9320, New Zealand
[2]National Institute of Water and Atmospheric Research, Wellington, New Zealand

*Correspondence to:* Jared Lewis
(jared@bodekerscientific.com)

**Abstract.** A method, based on climate pattern-scaling, has been developed to expand a small number of projections of fields of a selected climate variable ($X$) into an ensemble that encapsulates a wide range of indicative model structural uncertainties. The method described in this paper is referred to as the Ensemble Projections Incorporating Climate model uncertainty (EPIC) method. Each ensemble member is constructed by adding contributions from (1) a climatology derived from observations that represents the time invariant part of the signal, (2) a contribution from forced changes in $X$ where those changes can be statistically related to changes in global mean surface temperature ($T_{global}$), and (3) a contribution from unforced variability that is generated by a stochastic weather generator. The patterns of unforced variability are also allowed to respond to changes in $T_{global}$. The statistical relationships between changes in $X$ (and its patterns of variability) with $T_{global}$ are obtained in a 'training' phase. Then, in an 'implementation' phase, 190 simulations of $T_{global}$ are generated using a simple climate model tuned to emulate 19 different Global Climate Models (GCMs) and 10 different carbon cycle models. Using the generated $T_{global}$ time series and the correlation between the forced changes in $X$ and $T_{global}$, obtained in the 'training' phase, the forced change in the $X$ field can be generated many times using Monte Carlo analysis. A stochastic weather generator is used to generate realistic representations of weather which include spatial coherence. Because GCMs and Regional Climate Models (RCMs) are less likely to correctly represent unforced variability compared to observations, the stochastic weather generator takes as input measures of variability derived from observations, but also responds to forced changes in climate in a way that is consistent with the RCM projections. This approach to generating a large ensemble of projections is many orders of magnitude more computationally efficient than running multiple GCM or RCM simulations. Such a large ensemble of projections permits a description of a Probability Density Function (PDF) of future climate states rather than a small number of individual story lines within that PDF which may not be representative of the PDF as a whole; the EPIC method largely corrects for such potential sampling biases. The method is useful for providing projections of changes in climate to users wishing to investigate the impacts and implications of climate change in a probabilistic way. A web-based tool, using the EPIC method to provide probabilistic projections of changes in daily maximum and minimum temperatures for New Zealand, has been developed and is described in this paper.

# 1 Introduction

While future changes in climate will follow a single trajectory, it is highly unlikely that any single climate model projection will correctly simulate that trajectory. The use of a single model projection is therefore insufficient for assessing the potential future state of the climate. Rather, what is required is a large (e.g. 10,000 member) ensemble of projections that provides a probabilistic portrayal of how the climate is expected to evolve. Clustering of trajectories within that probabilistic envelope then shows where any single trajectory has a higher likelihood of occurring. Probabilistic simulations of future climate, presented as Probability Density Functions (PDFs), give decision-makers a much clearer picture of likelihoods of future climate states compared to a single projection, or a small set of projections (Watterson et al., 2008). That said, if decision-makers are presented with PDFs obtained from the same family of models, these may be biased by the assumptions and limitations inherent in a single family of models that do not explore the possible trajectories seen in other model families. PDFs of future climate that consider a greater number of sources of uncertainty, including uncertainty resulting from structural differences in the underlying models, provide more robust information needed for quantitative risk assessments, since the likelihood of any particular trajectory can be better estimated.

Exploring expected changes in extreme weather events also requires probabilistic simulations of future climate. While climate change may result in a small shift in the mean and/or standard deviation of a PDF of a selected climate variable, the tails of the distribution, which represent extreme weather events, can exhibit fractionally much larger changes (see Figure 1.8 in Solomon et al. (2007)). It is especially important that extreme events, which by their nature are unusual, are captured in an ensemble of projections.

Resolving changes in the frequency of regional-scale extreme weather events requires large ensembles of projections of high spatial and temporal resolution. Generating such ensembles using models which simulate all important physical processes, such as Global Climate Models (GCMs) or Regional Climate Models (RCMs), is currently computationally prohibitive. The ideas underlying climate pattern-scaling suggest a means of overcoming this hurdle and form the basis for the newly developed Ensemble Projections Incorporating Climate model uncertainty (EPIC) method described here. First, a robust statistical relationship is derived between the local climate variable of interest ($X$) and some associated readily generated predictor. In climate pattern-scaling, this predictor is typically the global mean surface temperature ($T_{global}$). If observations are being used to establish this relationship, then observed values of $X$ and $T_{global}$ would be used. If GCM or RCM output is used to establish the relationship, then $X$ and $T_{global}$ should come from the same model simulation.

Once the relationship between $X$ and $T_{global}$ has been determined, then, given multiple versions of $T_{global}$, multiple time series of $X$ can be generated based on that relationship. This methodology assumes that many versions of $T_{global}$ can be simulated in a way that captures the inherent variability resulting from structural uncertainties in GCMs and carbon cycle models in a computationally efficient way, e.g., through the use of a simple climate model (SCM). If the large ensemble of $T_{global}$ time series spans the range of model structural uncertainties, then the resultant ensemble of generated $X$ time series will reflect that spread in uncertainties, e.g., as done in Reisinger et al. (2010).

A number of previous studies (e.g. Murphy et al. (2007), Sexton (2012) and Harris et al. (2010)) used a method that was designed by the UK Met Office (Murphy et al., 2009) to provide probabilistic projections of future climate for Europe. Their method combines information from a perturbed physics ensemble (PPE), multi-model ensembles to capture model structural uncertainties, and observations. Since GCMs have been shown to not be structurally independant (Masson and Knutti (2011),

Knutti et al. (2013)), multi-model ensembles benefit from model weighting to improve the ensemble performance (Knutti et al., 2017). The limitations of these methods are that large computer resources are required to run the ensembles of simulations required which limit the ability to apply this method across many different greenhouse gas emissions scenarios.

## 2    Models and Data Sources

### 2.1    Regional Climate Model

An RCM simulation, or a number of RCM simulations, are used to provide the time series used to train EPIC i.e. to quantitatively establish the relationship between the change in annual mean global mean surface temperature and the change in the climate variable of interest and its variability. RCM simulations used in this study were performed using the Hadley Centre regional climate model HadRM3-PRECIS (Jones et al. , 2004) that has been modified to be used for New Zealand (Bhaskaran et al., 1999, 2002; Drost et al., 2007) and described in further detail in Mullan et al. (2016). The RCM domain spans 32°S to

52°S and 160°E to 193°E (167°W) on a regular rotated grid with a horizontal resolution of 0.27° and with the North Pole at 48°N and 176°E. Such a rotated grid, with the equator running through the New Zealand domain, ensures a quasi-uniform grid box spacing. The 0.27° resolution results in a domain of 75×75 grid points, reduces computation time for long simulations, and has been shown to be adequte in previous studies (Drost et al., 2007). The spatial resolution necessitates a computational time step of 3 minutes. The model orography and vegetation data sets were updated from those used by Drost et al. (2007) to

the high resolution surface orography data set used in NIWA's operational forecast model (Ackerley et al., 2012); differences in the vegetation fields are small. The first year of model simulation (the spin-up) is excluded from the analysis as this is used to achieve quasi-equilibrium conditions of the land surface and the overlying atmosphere.

The RCM lateral boundary conditions can be sourced either from meteorological reanalyses (these are typically used for hindcast simulations) or from Global Climate Model (GCM) output. The Atmosphere-only GCM (AGCM) used in this study

was HadAM3P developed by the Hadley Centre in the UK and forced by prescribed sea surface temperatures (SSTs) and sea ice extent at the air-sea interface for past and future climate simulations. HadAM3P is a slightly improved version of the atmospheric component of HadCM3 with 19 vertical levels and a horizontal resolution of 1.875° longitude by 1.25° latitude. HadAM3P simulate all atmospheric and land surface processes relevant to climate (Pope et al., 2000). Processes related to clouds, radiation, the boundary layer, diffusion, gravity wave drag, advection, precipitation and the sulfur cycle

are all parameterized in HadAM3P. Additional details regarding HadAM3P are available in Gordon et al. (2000), Pope and Stratton (2002), Pope et al. (2000), and Gregory et al. (1994). The output from the RCM was then statistically downscaled to a 0.05°×0.05° grid (Mullan et al., 2016).

The prescribed boundary conditions for the HadAM3P model were obtained from 6 Atmosphere-Ocean GCM (AOGCM) simulations obtained from the Coupled Model Intercomparison Project Phase 5 (CMIP5) archive, viz. simulations from the BCC-CSM1-1, CESM1-CAM5, GFDL-CM3, GISS-EL-R, HadGEM2-ES, NorESM1-M models. These AOGCMs were selected for their ability to best simulate changes in synoptic scale climate around New Zealand.

Most GCM and RCM simulations display biases when compared to observations. The RCM simulations used in this study were partially bias-corrected by bias-correcting the SSTs that are used as lower boundary conditions for the HadAM3P simulations, which then provided the lateral boundary conditions for the RCM simulations.

## 2.2 Simple Climate Model

In this study, MAGICC (Model for Assessment of Greenhouse-gas Induced Climate Change; Meinshausen et al. (2011); Mein-
shausen et al. (2011)) is the simple climate model (SCM) used to generate an ensemble of $T_{global}$ time series. MAGICC is a reduced complexity climate model with an upwelling diffusive ocean and is coupled to a simple carbon cycle model that includes carbon dioxide ($CO_2$) fertilization and temperature feedback parameterisations of the terrestrial biosphere and oceanic uptake. MAGICC can be tuned to emulate the behaviour of 19 different CMIP3 AOGCMs (Meehl et al., 2007) and 10 carbon cycle models (Friedlingstein et al., 2006). The resultant 190 different 'tunings' for MAGICC can be used to generate 190
equally probable $T_{global}$ time series that provide an indication of the spread in $T_{global}$ resulting from structural uncertainties in AOGCMs and the carbon cycle models used in C4MIP (Coupled Carbon Cycle Climate Model Intercomparison Project). When used as predictors for changes in local climate variables, and using the prior established quantitative relationship between $T_{global}$ and the $X$, these 190 $T_{global}$ time series can be used to generate 190 time series emulating $X$.

The EPIC method does not attempt to faithfully represent the full, true PDF of potential tuning parameters both for the
AOGCM tunings and the carbon cycle model tunings i.e. were MAGICC tuned to a different set of AOGCMs (e.g. the CMIP5 set rather than the CMIP3 set), we would obtain a different set of tuning files which could lead to a somewhat different spread in our generated ensembles. The purpose of this paper is not to generate perfect ensembles that encapsulate structural model uncertainty in a completely accurate way but rather to describe a method that provides a better representation of that uncertainty than can be achieved with only a limited set of RCM simulations. The robustness of the EPIC method depends on
the set of AOGCM and carbon cycle model tunings available and as more comprehensive sets (that better reflect the likelihood of some tunings over others) become available, we expect that the large ensembles generated by EPIC to better reflect the true underlying uncertainties.

## 2.3 Virtual Climate Station Network

While the RCM simulations have been partially bias corrected, we recognise that some biases may remain. Therefore, we build
our projections off an observational data set, so that, in the absence of any forced changes in climate, the projections default to observations (this is described in greater detail below). Observationally-based time series are obtained from the so-called Virtual Climate Station Network (VCSN). The VCSN data set for the New Zealand land surface is constructed on a regular $0.05° \times 0.05°$ grid from spatially inhomogeneous and temporally discontinuous quality controlled weather station data (Tait et

al., 2005). The values estimated on the $0.05° \times 0.05°$ grid are based on thin plate smoothing spline interpolation using a spatial interpolation model as described in Tait, (2008).

## 3 Methodology

For a given geographic location, each ensemble member, covering the period 1960 to 2100, is constructed from contributions including:

1. a climatology derived from observations that represents the time invariant part of the signal,

2. a contribution from long-term forced changes in the magnitude of the variable of interest where those changes scale with changes in anomalies in global mean surface temperature ($T'_{global}$), and

3. a contribution from weather, generated by a stochastic weather generator that incorporates both forced and unforced variability.

The construction of each of these signals is described in greater detail below with a high level overview of how these contributions are related shown in Figure 1. The methodology described below pertains to a selected single greenhouse gas (GHG) emissions scenario and the daily maps of the climate variable of interest ($X$; here daily maximum ($T_{max}$) and daily minimum ($T_{min}$) surface temperatures) are obtained from one or more RCM simulations. To produce the results for this study, 10 ensemble members were generated for each of the 190 $T_{global}$ time series from MAGICC to produce an ensemble of 1900 members. These ensemble members were generated over the period 1960 to 2100. The $T_{max}$ and $T_{min}$ fields were obtained from six RCM simulations driven by the Representative Concentration Pathway (RCP) 8.5 GHG emissions scenario for the period 1971-2100. RCP 8.5 was choosen as it displays a high climate signal to noise ratio, resulting in the most robust regression results (Huntingford and Cox (2000), Mitchell (2003)), but the methodology is valid for any choosen GHG emission scenarios assuming a robust regression fit is obtained during the training phase. The assumption, which has been verified (not shown here), is that the dependence of $X$ on $T_{global}$ is independant of the GHG emissions scenario used for the training. All anomalies were calculated with respect to the period 2000 to 2010. This anomaly period was choosen as the change in X over the $21^{st}$ century was of interest.

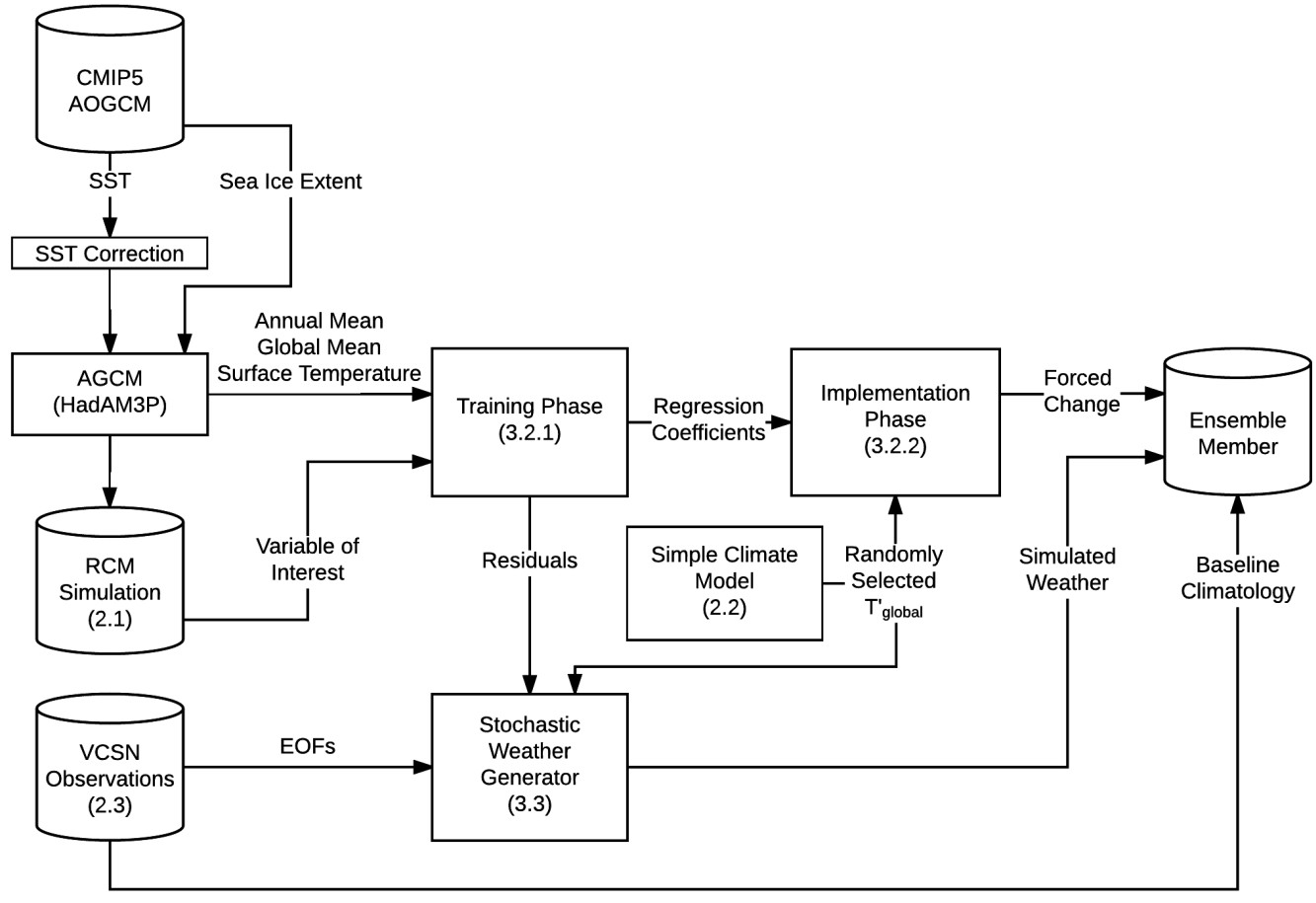

**Figure 1.** Flow chart illustrating the processes involved in generating a single EPIC ensemble member based on training from a selected RCM simulation. Numbers in brackets refer to the Section where more details are provided.

## 3.1 The climatology

At each $0.05°$ by $0.05°$ (approximately 5km) grid point, a mean annual cycle is calculated from daily observational data from 2001 to 2010. For this study, these observational data were obtained from VCSN (Section 2.3). Since the 10 year baseline period is rather short, a climatology derived by calculating calendar day means would still contain some weather-induced noise. Therefore, a regression model which includes an offset basis function, expanded in two Fourier series (Fourier pairs) to account for seasonality (see Section 2.4 of Kremser et al. (2014)), is fitted to the daily observational data to obtain the mean annual cycle. The first 2 Fourier series expansions are given in Eq. 1.

$$\beta(d) = \beta_0 + \beta_1 \times \sin(2\pi d/365) + \beta_2 \times \cos(2\pi d/365) + \beta_3 \times \sin(4\pi d/365) + \beta_4 \times \cos(4\pi d/365) \tag{1}$$

where $d$ is the day number of the year and $\beta$ is a the regression coefficient being expanded. By using an offset basis function expanded in Fourier pairs, the resultant mean annual cycle is smooth. Examples of the mean annual cycle are shown in Figure 2 for four selected locations around New Zealand.

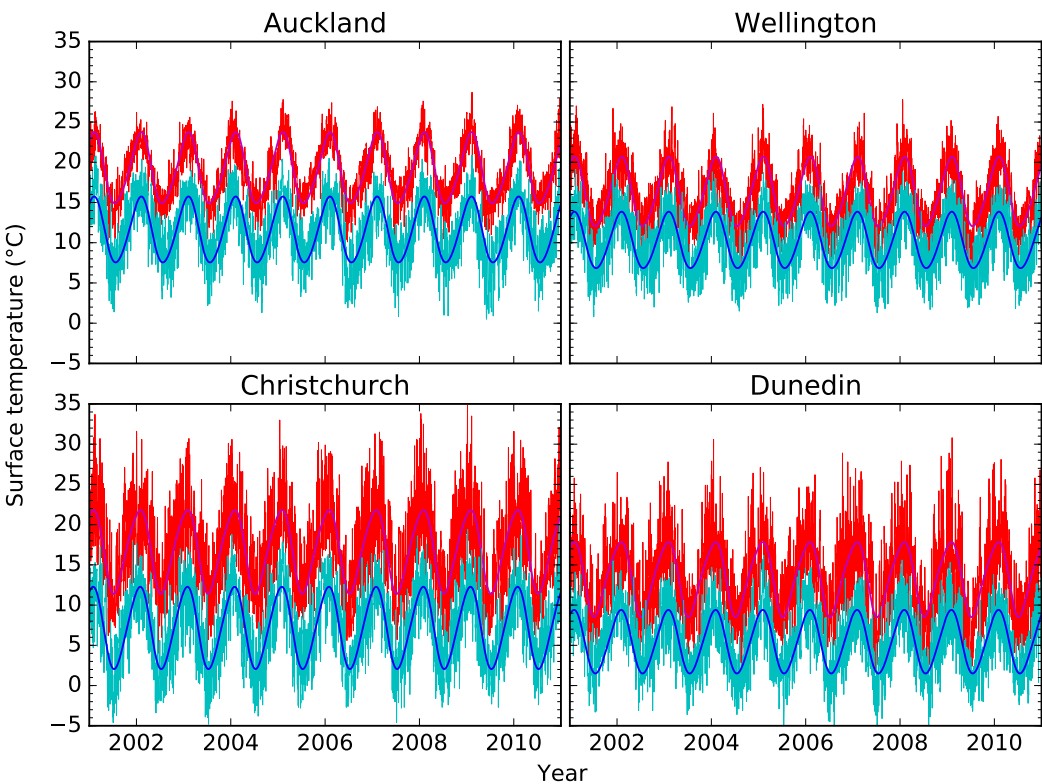

**Figure 2.** Observations of daily maximum surface temperature (red) and daily minimum surface temperature (cyan) from VCSN together with the mean annual cycle obtained from the regression model fit to the daily maximum surface temperature (magenta) and the daily minimum surface temperature (blue) time series for four selected locations in New Zealand, over the period 2001 to 2010.

This repeating mean annual cycle then provides the stationary baseline for the entire period of interest e.g. 1960 to 2100.

## 3.2 Direct response to $\mathrm{T}'_{global}$

### 3.2.1 Training Phase

In the training phase, the first-order long-term forced change in $X$ is established using the correlation between $X'$ and $\mathrm{T}'_{global}$. This relationship is expected to be dependent on the RCM simulation from which the variable of interest is obtained. There are two ways in which this can be managed, viz.:

1. A statistical relationship is quantified for each RCM simulation providing data for the training phase of EPIC. Then, in the 'implementation phase' of EPIC (see below), for each ensemble member, a single relationship is randomly selected.

2. A single statistical relationship is quantified using a concatenated time series obtained from all RCM simulations providing data for the training phase of EPIC. In the 'implementation phase', this relationship is used.

For the purposes of this study, method (1) is used as method (2) will tend to underestimate the true uncertainty of the relationship between $X'$ and $T'_{global}$.

A simple linear correlation between $X'$ and $T'_{global}$ is calculated for each of the 6 RCM simulations and each grid point independently, viz.:

$$X'(t) = \alpha \times T'_{global}(y) + R(t) \tag{2}$$

where $X'(t)$ are the daily anomalies with respect to the 2001-2010 mean annual cycle of $X$, $T'_{global}(y)$ are the anomalies of an annual mean global mean surface temperature time series obtained from the AGCM which provided the boundary conditions for the selected RCM simulation, $\alpha$ is the regression coefficient and R is the residual which is the part of the signal that cannot be explained by the statistical model. In this case, the residuals are used by the stochastic weather generator (see Section 3.3) to model higher order changes in the variability in $X$ which are not captured by Eq. (2).

The mean annual cycle of $X$, which is used to calculate $X'$, is generated using the same method and time period used to calculate the mean annual cycle of the observational set. $X'$, rather than $X$, is used in Eq. 2 as the change in the seasonal cycle is of interest. Removing the mean annual cycle removed the need to add additional terms to describe the baseline seasonal cycle.

Because GCM and RCM output provide a much longer time series than observations and extend into a period of greater changes in $X$, GCM or RCM output are preferentially used in this training phase. In this study, the input to the training phase of EPIC, $T'_{global}$ is sourced from the AGCM that provided the boundary conditions for the RCM simulation. Both the $T'_{global}$ time series used in the training phase and later in the implementation phase of EPIC need to be geophysically consistent. This geophysical consistency can be assessed by comparing the $T'_{global}$ time series obtained from the HadAM3P simulations, with the $T'_{global}$ time series obtained from the CMIP5 AOGCMs that provided the SST boundary conditions for the HadAM3P simulations (which were not used elsewhere in EPIC), as well as the 190 $T'_{global}$ time series obtained from MAGICC (Figure 3). There are clear differences between the $T'_{global}$ time series obtained from the CMIP5 AOGCMs and those obtained from the HadAM3P simulations. This is because the SSTs from the CMIP5 AOGCMs are bias corrected before being used as the surface boundary conditions for the HadAM3P simulations. The 6 $T'_{global}$ time series from the HadAM3P simulations (used in the training phase of EPIC) fall well within the envelope of the 190 MAGICC $T'_{global}$ time series used in the implementation phase of EPIC, even though MAGICC is emulating a range of CMIP3 models.

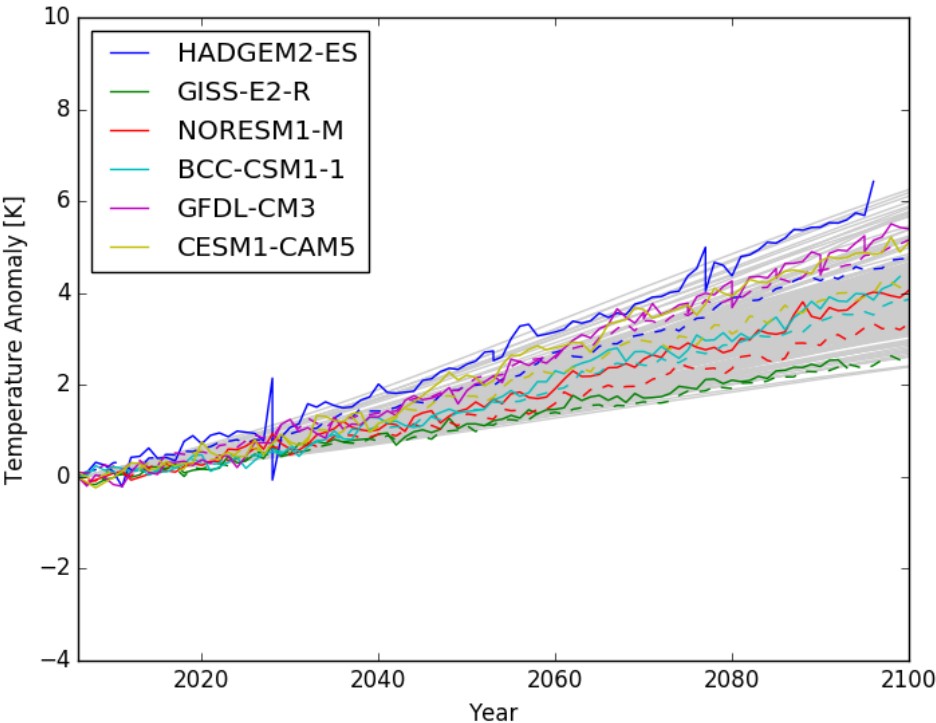

**Figure 3.** Annual mean global mean surface temperatures calculated from the CMIP5 AOGCM simulations under the RCP8.5 scenario (coloured solid lines). The annual mean global means from the HadAM3P (dashed lines) and MAGICC (grey lines) for RCP8.5 are also shown.

Because the fit coefficient, $\alpha$, is expected to depend on season, it is expanded in two Fourier pairs to account for its seasonality (Eq. 1). The resulting $\alpha$ has a smooth seasonal cycle which would not be the case if each month was fitted independently. When embedded in Eq. (2), the resulting Eq. (3), has five fit coefficients ($\alpha_0$ to $\alpha_4$).

$$X'(t) = (\alpha_0 + \alpha_1 \times \sin(2\pi d/365) + \alpha_2 \times \cos(2\pi d/365) + \alpha_3 \times \sin(4\pi d/365) + \alpha_4 \times \cos(4\pi d/365)) \times T'_{global}(y) + R(t) \quad (3)$$

5   The statistical model is solved using a multivariate least squares regression approach(Moore and McCabe , 2003) to obtain the fit coefficients. We refer to each such set of five fit coefficients as a tuple; recall that this fit is applied at each grid point and for each available RCM simulation.

An example of a fit of Eq. (2) to daily maximum surface temperature anomalies is shown in Figure 4 for a location in the South Island of New Zealand.

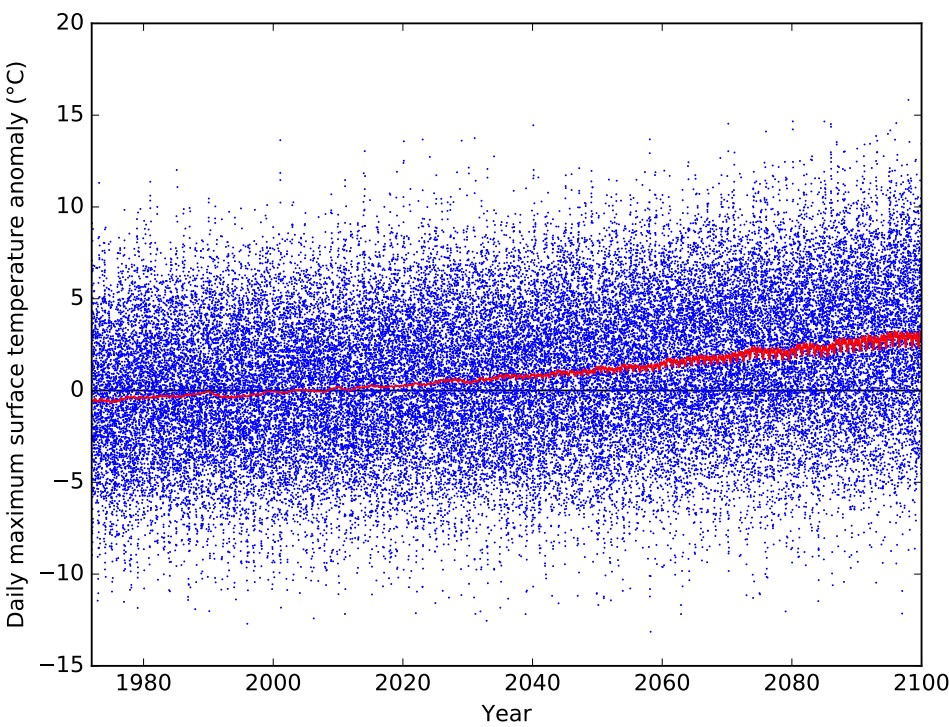

**Figure 4.** An example of the fit of Eq. (2) (red line) to daily maximum surface temperature anomalies (blue) obtained from the NorESM1-M RCM simulation under the RCP8.5 GHG emissions scenario at Alexandra, New Zealand (45.249°S, 169.396°E). Solid line represents the zero line (no change).

The small annual cycle in the fit, with growing amplitude, results from summer-time and winter-time daily maximum surface temperatures exhibiting different correlations against $T'_{global}$. The inter-annual variation arises from changes in $T_{global}$ as $\alpha$ does not change from year to year. In addition to the long-term forced change, there is significant day-to-day variability. The use of the residuals from such fits in the stochastic weather generator is described in Section 3.3.

5  The unitless $\alpha$ coefficient describes a location's sensitivity to changes in annual mean global mean surface temperature. The magnitude of $\alpha$ indicates whether $T_{max}$ or $T_{min}$ are changing faster ($\alpha>1$) or slower ($\alpha<1$) than the global mean surface temperature. Example maps of the $\alpha$ coefficient, over New Zealand, for four selected days throughout the year, are shown in Figure 5. This analysis shows that daily maximum surface temperatures over most of New Zealand are warming slower than $T_{global}$. However, high altitude regions, such as the Southern Alps, indicate $T_{max}$ increasing faster than $T_{global}$ for southern

10  hemisphere spring, summer and autumn.

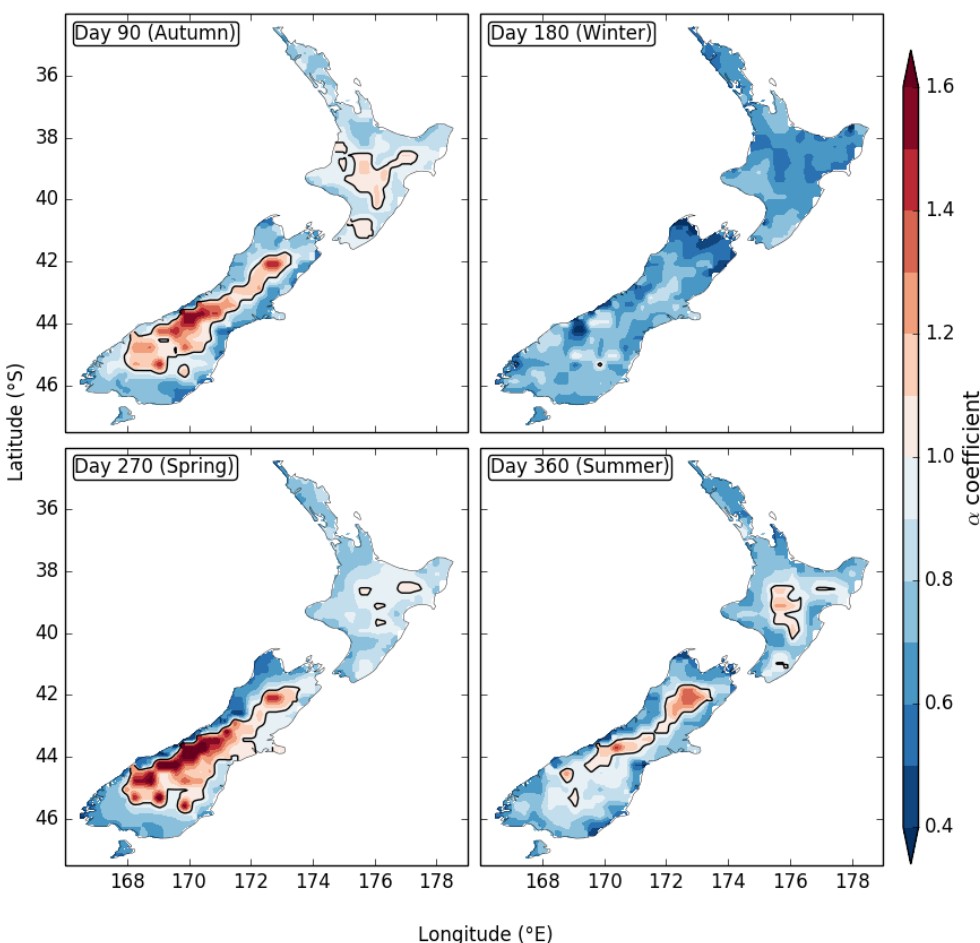

**Figure 5.** Maps of $\alpha$ coefficients (unitless) which represent the sensitivity of changes in daily maximum surface temperature to changes in annual mean global mean surface temperatures for locations throughout New Zealand. The $\alpha$ coefficients were derived from fits of Eq. (2) to daily time series of daily maximum temperatures at each grid point of the NorESM1-M RCM simulation. The annual mean global mean surface temperature anomalies were taken from the AGCM simulation that provided the boundary conditions for this particular RCM simulation. Black lines indicates $\alpha$ values of 1.0.

There is, of course, some uncertainty in $\alpha$. To account for that uncertainty, a large set of $\alpha$ tuples is derived through a Monte Carlo bootstrapping approach (Efron and Tibshirani , 1994), whereby residuals from the Eq. (2) fit are randomly sampled and added to the regression model fit to generate multiple statistically equivalent time series which are then refitted to obtain equally probable $\alpha$ fit coefficients (Bodeker and Kremser, 2015). This approach allows for the incorporation of the uncertainty in the fit of Eq. (2) into the final ensemble of projections.

### 3.2.2 Implementation Phase

Once the Monte Carlo derived sets (just one set if method (2) is used) of $\alpha$ tuples have been obtained, they are used in the implementation phase of EPIC. As described in Section 2.2, 190 simulations of $T'_{global}$ can be generated using a SCM. A randomly selected $T'_{global}$ time series from the 190 member set is used together with a randomly selected tuple of $\alpha$ values to
generate a series of maps of $X'_{forced}$ using Eq. (2), where the forced subscript denotes these are changes which correlate with $T'_{global}$

There might be some concern that the random selection of an $\alpha$ tuple from the available set of tuples for a location could cause the spatial coherence in the forced signal across New Zealand to be lost, as at a nearby location a different tuple could be randomly selected. This was tested for and was found not to be the case as the multiple instances of tuples (multiple instances
of Figure 5) are very similar and consistent (not shown here).

### 3.3 Indirect response to $T'_{global}$ and weather noise

In addition to the change in $X$ that correlates directly with $T'_{global}$, higher order components of variability, as well as realistic weather noise, must be present in the projections comprising the ensemble. One potential use of the ensemble of projections generated by EPIC is assessment of the impacts and implications of climate change on a regional scale. These impacts seldom
happen at a single site, i.e. the impact is felt over a large area. For this reason it is important that any specific member of an ensemble is appropriately spatially coherent over multiple sites. This is not achieved if the method considers each site in isolation since any purely stochastically determined weather noise added to a site would not be spatially coherent at neighbouring sites. For this reason, an empirical orthogonal function (EOF) approach, described by Lorenz (1956), is used to describe the spatial weather patterns and how they change over time. EOF analysis is a statistical method which reveals the spatial patterns,
or modes of variability in a data set, and how these patterns evolve over time as given by the resulting principal component (PC) time series. Hereafter we refer to these modes of variability as 'weather modes'. The EOF analysis is applied to $X'$ after the dependence on $T'_{global}$ has been removed. These weather modes, and PC time series, are then used to construct a weather generator which produces realistic weather noise by stochastically generating PC time series ($PC_{syn}$). The following is recognised in the construction of the stochastic weather generator:

1. That VCSN data will provide the most realistic representation of weather noise.

2. That RCM simulations will simulate how that weather noise is likely to evolve in response to climate change (represented by $T'_{global}$).

3. That the RCM simulations will be imperfect in simulating the patterns of variability derived from the VCSN data.

4. That there will be patterns of variability (weather) whose amplitude and variability will respond to climate change as
well as others which do not change with increases in $T'_{global}$.

### 3.3.1 Identifying the modes of variability responding to climate change

We begin by conducting an EOF analysis on VCSN data that have been detrended by removing the first order trend and on residuals from the fit of Eq. 2 to RCM data in the training phase. Where the patterns of variability obtained from EOF analyses of VCSN and RCM diverge is considered to be the cut-off point for where the RCM simulation can be taken to have any integrity with regard to simulating forced changes in weather noise. Visual inspection of the EOF maps derived from VCSN and RCM data suggested that the first four modes of variability are well represented by the RCM simulations (see Figure 6).

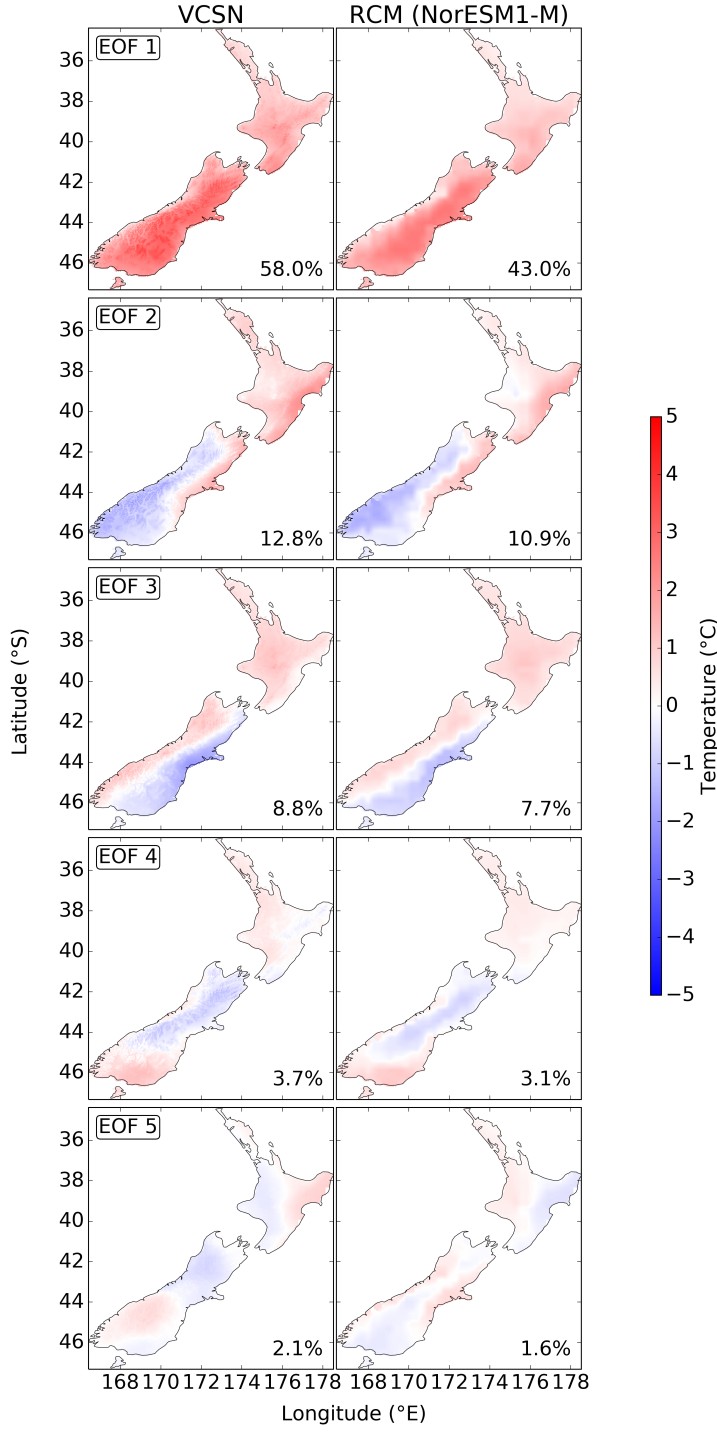

**Figure 6.** The first five EOF patterns of weather noise in daily maximum surface temperatures obtained from VCSN data from 1972 to 2013 (left column) and obtained from RCP8.5 NorESM1-M RCM output from 1972 to 2100. The colour bar shows the amplitude of the pattern in °C. The percentage values in each panel show the fraction of the total variability explained by each mode.

It is clear from Figure 6 that the RCM EOFs exhibit the same modes of weather variability as seen in the VCSN data up until EOF pattern 4. Together, the first four patterns of variability explain 83.3% of the total weather variability in the VCSN data and 64.7% of the variability in the RCM data. It is these four modes of weather variability that evolve with $T'_{global}$ in our stochastic weather generator.

### 3.3.2 Modelling forced changes in the amplitude and variability of weather modes

To compare statistics from the PC time series calculated from VCSN and RCM data, they must share the same underlying weather modes. This is done by projecting the VCSN weather modes ($EOF_{VCSN}$) onto the RCM data to calculate a pseudo-PC time series. A pseudo-PC time series is calculated in the same way that a standard PC time series is calculated, except that the weather modes are prescribed instead of being calculated from the data. A pseudo-PC time series describes the magnitude of a particular pattern of variability from VCSN, which is present in the RCM data. The VCSN weather modes, rather than the RCM weather modes, were prescribed because the observational data set is more likely representative of patterns of variability seen in New Zealand. The pseudo-PC and $VCSN_{PC}$ time series can be compared as they both describe the same patterns of variability.

In the stochastic weather generator, we consider changes in:

1. The amplitude of the weather mode: this is quantified by correlating the associated pseudo-PC time series with $T'_{global}$ and then using that correlation coefficient ($\beta$) to drive a trend in the PC time series obtained from the VCSN-based EOF analysis.

2. The variability of the weather mode: this is quantified by correlating the variability in the associated pseudo-PC time series with $T'_{global}$ and then using that correlation coefficient ($\beta_{var}$) to drive a trend in the variability of the PC time series obtained from the VCSN-based EOF analysis. The mean variability of the weather mode is obtained from the VCSN PC time series rather than the pseudo-PC time series, so that the weather mode emulates the magnitude of variability seen in the VCSN data.

We also recognize that the PC time series will exhibit temporal auto-correlation and therefore that correlation is quantified and removed before correlating the PC signal, and its variability, against $T'_{global}$. The resulting time series ($PC_{syn\_n}$) captures both long-term shifts and/or changes in spread of the $n^{th}$ weather mode. We note, however, that by considering only lag-one autocorrelation in these PC time series, we neglect longer term auto-correlation, e.g. that resulting from El Niño and La Niña events. As a result, our ensemble time series exhibit smaller inter-annual variability than is observed in VCSN time series.

The ability of the method to generate a set of PDFs of the $PC_{syn\_1}$ to $PC_{syn\_4}$ time series is demonstrated in Figure 7.

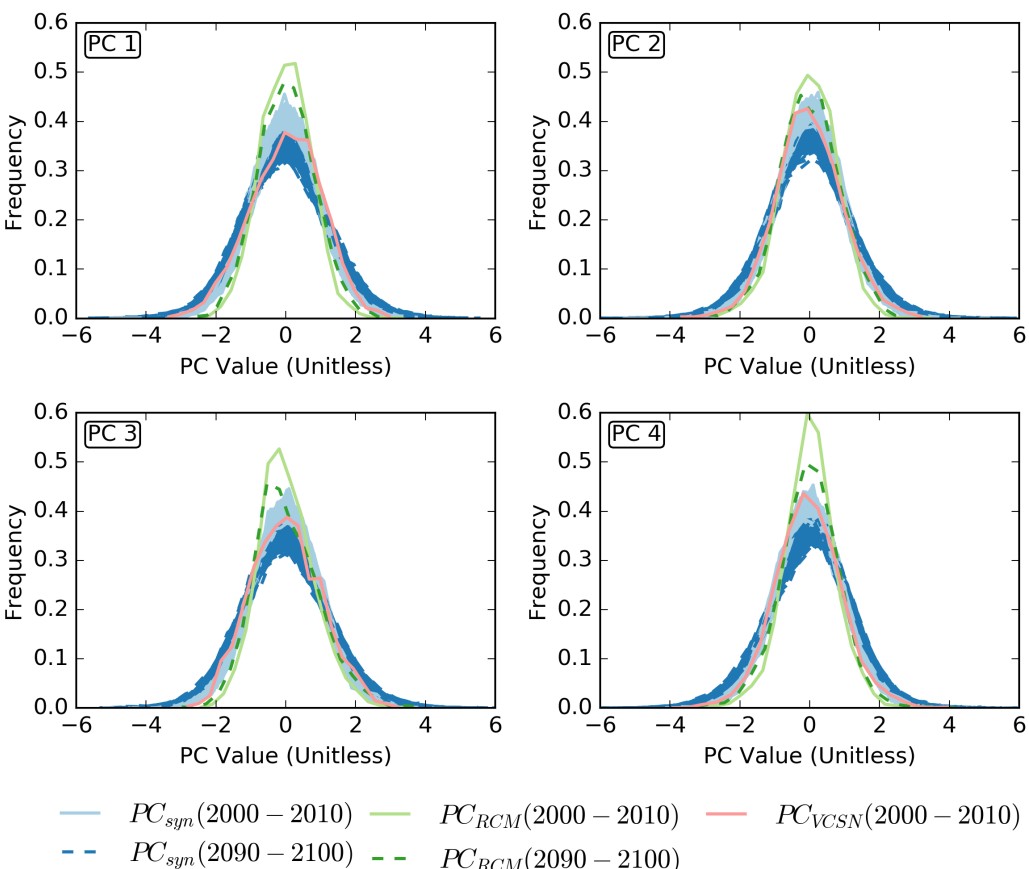

**Figure 7.** PDFs of the first four synthetic (PC$_{syn}$) and RCM (PC$_{RCM}$) PC time series for the first decade of the $21^{st}$ century are shown in solid lines and PDFs for the last decade of the $21^{st}$ century are shown in dashed lines. PC$_{RCM}$ and PC$_{syn}$ were both derived from the NorESM1-M RCM output as an example. The PDF from the PC time series (2000-2010) obtained from VCSN is also shown (PC$_{VCSN}$). The disagreement between the PC$_{RCM}$ and PC$_{VCSN}$ validates the use of VCSN weather noise as the basis for our stochastic weather generator and the good agreement between the ensemble of PC$_{syn}$ and PC$_{VCSN}$ demonstrates that the EPIC method generates synthetic PC time series with a degree of variability that matches reality.

The EPIC method corrects for any shortcomings in the ability of the RCM to correctly simulate expected magnitudes of weather variability for these four primary modes and then accommodates these corrections when generating PC time series that evolve into the future.

### 3.3.3 Modelling higher order modes of variability in weather

5 The stochastic weather generator includes the effects of EOF patterns five and higher but assumes that these modes show no dependence on $T'_{global}$ as the RCM simulations do not accurately simulate these higher modes of weather variability. The variability of the PC time series often has a strong seasonal cycle. Therefore, for EOF pattern five and higher, synthetic PC time series ($PC_{syn}$) are generated using a standard Monte Carlo approach, i.e. randomly selecting values from $N(0, \sigma(d))$,

that is, a normal distribution with a mean of 0 and a standard deviation which depends on the day of the year which is being modelled. $\sigma(d)$ is determined by a linear least squares fit of two Fourier pairs (Eq. 1) to the VCSN PC time series. The Fourier pairs model the seasonal cycle in the PC time series. This approach allows selection of extreme PC values that are outside of the range of PC values experienced in the 1972-2013 period, but noting that the PDFs of these PCs do not evolve with time.

As with the forced changes in the amplitude and variability of weather modes, the auto-correlation in the PC time series is also quantified and captured in the statistically modelled PC time series.

For a given ensemble member, once synthetic PC time series at daily resolution have been generated, they are used to produce a reconstructed weather field, $W$, according to:

$$W(i,j,t) = \sum_{n=1}^{50} EOF_{VCSN\_n}(i,j) PC_{syn\_n}(t)$$

where i, j and t represent the latitude, longitude and time dimensions respectively and n is the nth weather mode.

Since $W$ has been constructed from a linear combination of spatial patterns of variability, each of which is spatially coherent, it retains the property of spatial coherence. The variability evolves as expected under changes in $T'_{global}$ for the first four modes of variability, as simulated by the RCM, and where extreme conditions, outside the range of the training period, do occur with a statistically reasonable frequency due to the stochasticity in the construction of the pseudo PC time series.

$T_{min}$ is modelled identically to $T_{max}$ with one small change: days with anomalously low $T_{max}$ would be more likely to have anomalously low $T_{min}$. Not accounting for this correlation could result in stochastically modelled $T_{min}$ values being higher than the modelled $T_{max}$ value for that day. To avoid that, and to capture the correlation between $T_{max}$ and $T_{min}$ on any given day, the same set of random numbers used to generate the values in the synthetic $PC_n$ time series for $T_{max}$ for a given day is used to generate the values in the synthetic $PC_n$ time series for $T_{min}$. This forces the selection of $PC_{syn}$ values from the

same region of the PDF for both $T_{max}$ and $T_{min}$.

## 4   Results

Examples of the $T_{max}$ and $T_{min}$ time series generated by the EPIC method are shown in Figure 8 for four population centres in New Zealand together with the associated VCSN time series.

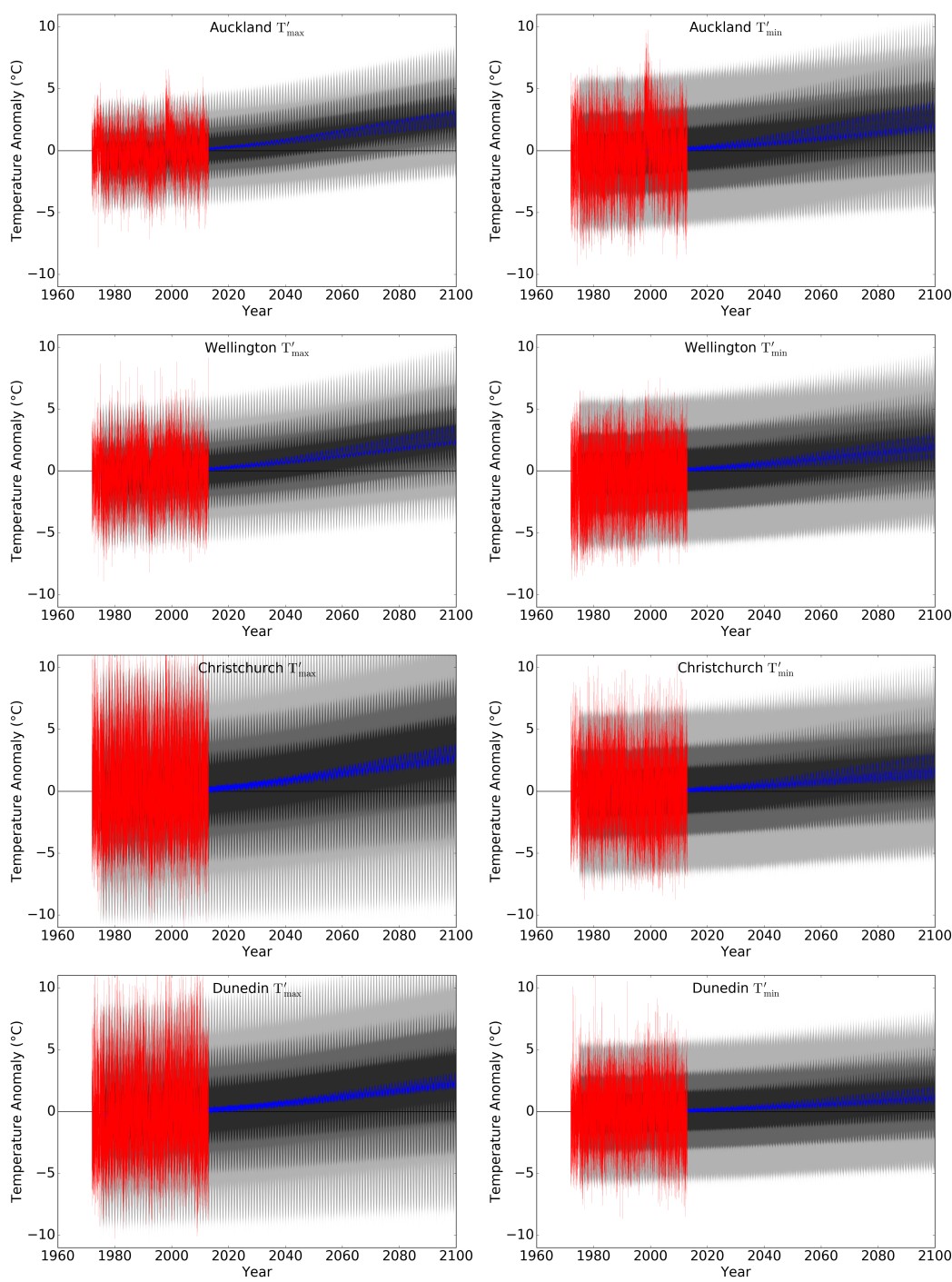

**Figure 8.** Example output from 1900 EPIC-generated time series for Auckland, Wellington, Christchurch and Dunedin from 1960 to 2100 under the RCP8.5 GHG emissions scenario. Grey shaded areas show the 1, 10, 25, 75, 90 and 99 percentiles while the blue line shows the median value on each day. $T'_{max}$ (left column) and $T'_{min}$ (right column) anomalies with respect to the 2000-2010 mean annual cycle. VCSN time series are overlaid in each panel (red lines).

Actual EPIC ensemble time series add these anomaly time series to the 2001-2010 VCSN-derived annual cycle climatology and therefore show no systematic bias with respect to the VCSN data. The EPIC-generated time series also show a long-term evolution consistent with expectations from RCM simulations, including the effects of the spread in those simulations. While it cannot be directly seen from the time series plotted in Figure 8, the EPIC-generated time series also exhibit changes in weather variability consistent with RCM projections of expected changes in the first four modes of weather variability. The apparent annual cycle in the anomaly time series reflects the annual cycle in the variance and not an annual cycle in the anomalies; towards the end of the period there is a true annual cycle in the anomalies from differential seasonal changes in $T_{max}$ and $T_{min}$. The inter-annual variability of the EPIC ensemble members is lower than that of the observational data set. This is due to EPIC not including any terms which describe patterns of variability which occur at time scales of longer than 1 year.

## 5   Discussion and Conclusions

The EPIC (Ensemble Projections Incorporating Climate model uncertainty) method, is able to generate large ensembles of daily time series of daily maximum and minimum temperatures that exhibit the following characteristics:

- No bias with respect to VCSN data.

- Long-term evolution consistent with projections from a suite of RCM simulations, incorporating the uncertainties inherent in those simulations as well as additional structural uncertainties that may arise from the use of a wider suite of RCMs as captured by the use of projections of $T'_{global}$. $T'_{global}$ time series were generated by a SCM tuned to 19 different AOGCMS and 10 different carbon cycle models and used as a predictor for the long-term change in $T_{max}$ and $T_{min}$.

- Weather variability with extremes that extend beyond that observed in the VCSN record and that evolve in a way consistent with RCM projections of changes in the four primary modes of weather variability.

- Spatial coherence in weather variability in any single ensemble member is preserved.

As such, EPIC-generated projections are suitable for generating robust PDFs of projections of $T_{max}$ and $T_{min}$.

The number of members in each ensemble is essentially limited only by the computing resources available. The stochasticity introduced by the Monte Carlo analysis and modeling of the weather noise allows for many ensemble members to be generated for a given $T_{global}$. For calculating the PDFs that are delivered to users, we currently generate 19,000 member ensembles (10 ensemble members for each $T_{global}$) for a given RCP scenario at each $0.05° \times 0.05°$ grid point across New Zealand.

A web-based tool has been developed to deliver PDFs of $T_{max}$ and $T_{min}$ for the period 2001-2010 and 2091-2100 to users along with statistics regarding the change in frequency of extreme events, i.e. days per year with $T_{max}$ above $25°C$ and $30°C$ and $T_{min}$ below $0°C$ and $2°C$. The tool is available at http://futureextremes.ccii.org.nz/.

The next steps for the development of EPIC include extending the range of climate variables to daily surface broadband radiation, surface humidity and precipitation, and incorporating longer term sources of variability, e.g. those generated by El Niño and La Niña events, into the stochastic weather model. The implementation of a model weighting scheme, such as Knutti et al. (2017), for the training data could increase the applicability of the model.

## 6 Code and data availability

The source code and data used is available upon request to the corresponding author. The VCSN data set employed is available from NIWA (https://www.niwa.co.nz/climate/our-services/virtual-climate-stations).

*Acknowledgements.* This research was funded by the Ministry of Business, Innovation and Employment as part of the Climate Changes,
5  Impacts and Implications programme (contract C01X1225). We would also like to thank Malte Meinshausen for providing the tuning files used to emulate various AOGCMs using MAGICC, and Abha Sood for assistance with the RCM data.

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
