# Peer review of "A method to encapsulate model structural uncertainty in ensemble projections of future climate: EPIC v1.0"

_Geoscientific Model Development, 2017_

## Short Comment (SC1) · 4 Apr 2017

Dear authors,

in my role as Executive editor of GMD, I would like to bring to your attention our Editorial version 1.1:

http://www.geosci-model-dev.net/8/3487/2015/gmd-8-3487-2015.html

This highlights some requirements of papers published in GMD, which is also available on the GMD website in the 'Manuscript Types' section:

http://www.geoscientific-model-development.net/submission/manuscript_types.html

In particular, please note that for your paper, the following requirements have not been met in the Discussions paper:

- "The main paper must give the model name and version number (or other unique identifier) in the title."

In the abstract you provide an acronym (EPIC) for your newly developed method. In order to simplify reference to your developments, please add this acronym and a version number (you might like to improve / bugfix it in the future ) in the title of your article in your revised submission to GMD.

Yours,

Astrid Kerkweg

---

## Referee Comment (RC1) · Anonymous Referee #1 · 8 May 2017

In this paper the authors have developed a method to encapsulate structural uncertainty in ensemble projections of future climate by combining regional climate model output with that from a simple climate model. The aim is produce a large ensemble of climate variables representing that which would be produced from a global climate model if it weren't made impossible by computational demands. A small ensemble of climate variables (T_min and T_max) and annual mean global temperature is produced from the regional climate model. The relationships between the climate variables and temperature are used to produce a large ensemble of the climate variables from the more readily available annual mean global mean temperature (simulated in the simple climate model). Observations are used to provide climatology, weather variability and maintain spatial coherence to the predictions. I think the underlying method that has been developed is interesting and would benefit the scientific community if published,

my concern about publishing in GMD is that this paper is about a method rather than a model. If the method has been developed such that it is a model in itself then the paper needs to be focussed on this and it requires more discussion about how users will run the model beyond what is presented in the paper, particularly with the new climate variables that will be included, how a new region is considered, how a new RCP is used and what will happen to the model when developments to the simple climate model and global/regional climate model are made. It may perhaps be better placed in a journal such as 'Advances in Statistical Climatology, Meteorology and Oceanography (ASCMO)'.

I will continue the review based on the method development.

General comment:

As a method I think this is a neat way to produce a large ensemble of climate variables at a regional level that would otherwise be unavailable and attempting to encapsulate the structural uncertainty and weather variability. In some places I find the method hard to follow and the paper needs clarity. In some places I think it would benefit from further equations to back up the text. I would also like to see more discussion on how parametric uncertainty could be included or why it is not deemed necessary. I would also like to see how additional RCPs are included in the method – are they included in the final PDFs or do you envisage separate PDFs for each scenario?

Particular points:

1. Introduction:

There is not enough of a literature review here. In particular, the UK Met Office have developed methods to produce probabilistic climate projections with a similar aim to this paper. Please discuss how this work achieves the goal differently. It would be good to see the work of Tebaldi and Knutti referenced here too discussing the difficulties in producing probabilistic information from multiple models.

In particular, the following papers should be referenced and discussed. Murphy, J. M. et al. A methodology for probabilistic predictions of regional climate change from perturbed physics ensembles. Phil. Trans. R. Soc. A 365, 1993–2028 (2007). Sexton, D., Murphy, J., Collins, M. & Webb, M.Multivariate predictions using imperfect climate models: Part 1 outline of methodology. Clim. Dynam. http://dx.doi.org/10.1007/s00382-011-1208-9 (2011). Harris, G. R., et al. "Probabilistic projections for 21st century European climate." Natural Hazards and Earth System Sciences 10.9 (2010): 2009-2020.

Line 2: 'will correctly simulate that trajectory'. I think the inclusion of 'correctly' is necessary since all climate models are attempting to simulate this trajectory.

2.1 Regional Climate Model

How dependent on the RCM are the relationships that are established? Would the relationships be expected to change with a new RCM?

Page 3, Line 10: What does 'adequate' mean? Page 3, Line 21: 'The model simulates all atmospheric and land surface processes'. There are missing processes and those that are included are subject to uncertainties – is the method robust to this? Page 3, Line 26: What is the implication of the remaining bias?

2.2 Simple climate model

Is MAGICC the only such model? How is it known to simulate annual mean global mean temperature adequately? Are the 19 AOGCMs and 10 carbon cycle models defined by MAGICC or could a user change what is included?

Page 4, Line 2: It is you that considers the 190 simulations as equally probable. 'We consider the resultant 190 different 'tunings' for MAGICC to be equally probable.' Could a user make different decisions? Page 4, Line 3: What does 'some' mean? Since the title of the paper states that the structural uncertainty is encapsulated this needs explaining. Page 4, Line 7: This is the first mention of 'local' - I think this should be

included earlier as it's an important point. Earlier, X is used. It would add clarity to define X better earlier on and use X here and in future.

2.3 Virtual Climate Station Network

Is this the best observational dataset beyond New Zealand?

3. Methodology

This section would benefit from some equations to clarify the text – especially right at the beginning to go with the bullet points.

Why is the period 1960-2100 used?

Page 4, Line 27: 'one or more RCM simulations' – I don't quite understand how the RCM simulations are being used here? Later it says you use five – what is the benefit of five and how did you choose these five? How could a user choose a different number? Page 4, Line 28: You haven't defined alpha yet so this is confusing. Can you state what the sampling of these alpha values is representing instead? Page 5, Line 2: 'valid for any GHG emissions scenarios' – is it valid if the regression isn't robust? How can a user validate this if they change the GHG scenario? Page 5, Line 2: Why is the anomaly calculated to the 2000-2010 baseline? You later say it is rather short so please say why you have chosen it.

3.1 Climatology

Page 5, Line 9: Are you following the method cited or have you expanded it? Can you say why you have chosen this particular method to account for seasonality? Figure 2: I think magenta and red are mixed up.

3.2.1 Training phase

I find this section particularly hard to follow and think it would benefit from more precise equations. Equation 2 doesn't help to explain what you actually did in terms of using time series from 5 RCMs. How have you built a model to explain the relationship between variables of a different scale (daily and annual)? How robust is this relationship with only 5 RCMs? Can you explain the statistical models that you build and how they are validated to produce robust relationships that can be used with the simple climate model output?

Page 7, Line 6: The relationship itself is calculated over the period 1960-2100? Page 7, Line 7: 'an annual mean global mean ..temp.. series' – do these match the period of X'? Are there five series from the five models? Page 7, Line 7: $\alpha$(d) is the fit coefficient – are there 365 of these values? Page 7, Line 13: GCM 'and' RCM. Page 7, Line 17: I don't see how the Fourier pairs are embedded in equation 2 to find five fit coefficients from 365? Page 7, Line 18: 'recall, that this is applied at.' – I don't see where it told me that is is applied for each RCM simulation – are they not all used to find the relationship? How do you choose a specific RCM simulation? You should also remind the reader at the start of this section that this is done at a grid box level. Figure 3: Could you zoom in to show what the red line looks like? It's quite hard to see. Equation 2 must be more complicated than it looks to produce this line. Page 8, Line 5: 'The unitless $\alpha' - \alpha$ is five values obtained from 365 (maybe more for the five RCMs) so I can't picture what it really is or how it represents sensitivity to temperature. I'm sorry I'm quite lost with respect to alpha. Maybe more equations would help. Page 9, Line 6: alpha depends on the RCM. How have you used the RCMs here? Have you chosen one – if so, why start with five? What happens to the data from the other 4? Have you found alpha with all five and then randomly selected from all of them to produce the MC sample? Perhaps reordering the writing here might clear things up with regards to how you are using the RCMs.

3.3 Indirect response. . ..

Page 10, Line 26: What does series mean here? Please be more precise. Page 11, Line 4: I don't understand this point. Can you reword it to be more precise.

3.3.1 Identifying the modes. . .. . .

Page 11, Line 6: Are you referring to the residuals in Figure 3? Your language appears to have changed here and it sounds like you are doing something new. If these are the residuals please use the same language and link it better to the previous sections. Page 11, Line 7: 'Where the patterns of variability obtained from EOF projections of VCsN. . ..' Figure 5: Can you interpret these EOFs? Why is the period 1972-2013 used here? Also, this is not the RCM discussed in the introduction? Why have you changed the RCM?

3.3.2 Modelling forced changes. . ..

Page 13, Line 12: Is it just New Zealand or is it likely to be more representative everywhere? Page 13, Line 14: Equations would be help clarity here. I'm still struggling to understand how you are correlating data on different scales. Page 13, Line 23: How did you remove the autocorrelation? Page 13, Line 25: Why did you not remove autocorrelation at larger time lags? Can you talk about the implications? If it would matter that the interannual variability is too small why not calculate and remove it? If it's not 'too small' then justify not doing it. Page 13, Line 28: How did you create pdfs from time series? Is the data from the whole time series in the pdf or are these time slices? What is the implication of having time series data in these pdfs? Have you got a sample of PCs to create the pdf?

3.3.3 Modelling higher order modes. . ..

Page 15, Line 2: Can you better explain where ïĄş comes from? Perhaps use an equation. Page 15, Line 9: Why is n=50? Figure 7: Can you show a close up of one of the blue lines perhaps – it's difficult to see how the smaller scale temporal patterns are captured and discussed later. Page 17, Line 5: I don't understand the line starting 'The apparent annual cycle. . .'. Could you elaborate? Is this what you'd expect and is it adequate? Page 17, Line 9: Are there any implications to the interannual variability being smaller? Is improving this a future direction? Page 17, Line 23: Can you say where 19000 comes from? Previously, you mentioned 1900 simulations so I'm unsure

what the extra simulations are taking into account.

Technical points: Abstract: Page 1, Line 11: change the direction of the first quote mark. This happens in other parts of the paper too. Introduction: Page 2, Line 7: Do you mean 'uncertain'? Page 4, Line 28: Make all of global subscript – use _{global}.

---

## Referee Comment (RC2) · Anonymous Referee #2 · 9 May 2017

Jared and colleagues present a method to compute distributions of local weather variables, and provide an example of how it can be applied to the case of New Zealand. For each geographic location, the method generates combinations of local climatology, long-term forced changes, and stochastic weather. By combining long-term temperature projections consistent with a large set of AOGCMs and carbon-cycle models, the paper claims to encapsulate model structural uncertainty in ensemble projections of future climate, and climate change.

The method proposed by the paper is interesting. However, the paper neglects references to earlier literature that have proposed related approaches. Furthermore, the method as it is currently described shows some serious shortcomings, particularly in the assessment and inclusion of structural uncertainty in the Probability Density Functions (PDFs) suggested. Two issues stand out:

1) Rather than encapsulating structural model uncertainty in a sensible and robust way, the current method basically multiplies and preserves model sampling bias. Just like the proposed method explores stochastic weather variations with an EOF analysis to understand dominant modes of variability, the same should be carried out for the 19 AOGMs and carbon-cycle models. The implicit assumption that each AOGCM realization is statistically or structurally independent is not supported. This would benefit strongly from appreciating the findings from Masson & Knutti (2011) or Knutti et al (2013) to determine the structural independence of AOGCMs, and apply the methods of model weighting as described in Knutti et al (2017) in order to address structural model uncertainty in a meaningful way.

2) The proposed method hinges on the assumption that fields of variable X are independent of the structural model uncertainty in AOGCMs. This assumption is not supported by any evidence. Not all patterns have to be equally probable to occur with a certain T_global response to a specific forcing path. What is required here is evidence based on a re-analysis of AOGCM data which shows that local patterns (or patterns of boundary conditions for the RCM) are either equally probably across high and low-response AOGCMs, or differ across these responses pointing towards the limitations of the proposed method.

The claims about the applicability and usefulness of the method would be unsupported if both these points are not addressed in a significantly revised manuscript.

Smaller editorial comments:

P4L28: T_global is formatted incorrectly

P5L2: Please edit this sentence for spelling and grammar. The authors need to provide evidence to make the claim that the methodology is valid for any chosen GHG emissions scenario.

P6Fig2: Color descriptions do not match the figure.

References:

Masson, D., and R. Knutti (2011), Climate model genealogy, Geophys. Res. Lett., 38, L08703, doi:10.1029/2011GL046864.

Knutti, R., D. Masson, and A. Gettelman (2013), Climate model genealogy: Generation CMIP5 and how we got there, Geophys. Res. Lett., 40, 1194–1199, doi:10.1002/grl.50256.

Knutti, R., J. Sedláček, B. M. Sanderson, R. Lorenz, E. M. Fischer, and V. Eyring (2017), A climate model projection weighting scheme accounting for performance and interdependence, Geophys. Res. Lett., 44, 1909–1918, doi:10.1002/2016GL072012.

---

## Author Comment (AC1)

**Authors' response to the Referees**

For clarifying our answers to the reviewers' comments, the following scheme is used: Comments of the reviewers are denoted in blue while the authors response is given in black. The changes made to the manuscript in response to the reviewers comments have been appended.

It should also be noted that the title of the paper has been changed to *"A method to encapsulate model structural uncertainty in ensemble projections of future climate: EPIC v1.0"* in reference to Short Comment #1.

**Anonymous Referee #1**

We would like to take this opportunity to thank the reviewers for their thorough review of this paper. We recognize the value of their time and appreciate the improvements in our paper that this review has led to.

In this paper the authors have developed a method to encapsulate structural uncertainty in ensemble projections of future climate by combining regional climate model output with that from a simple climate model. The aim is produce a large ensemble of climate variables representing that which would be produced from a global climate model if it weren't made impossible by computational demands. A small ensemble of climate variables (T_min and T_max) and annual mean global temperature is produced from the regional climate model. The relationships between the climate variables and temperature are used to produce a large ensemble of the climate variables from the more readily available annual mean global mean temperature (simulated in the simple climate model). Observations are used to provide climatology, weather variability and maintain spatial coherence to the predictions. I think the underlying method that has been developed is interesting and would benefit the scientific community if published, my concern about publishing in GMD is that this paper is about a method rather than a model.

We agree that the paper is more about a method than a model but it is a method that extends the utility of regional climate models and so we felt that it would still be of considerable interest to the modelling community. We therefore felt that GMD would still be a suitable vehicle for communicating this method to the target audience.

If the method has been developed such that it is a model in itself then the GMDD paper needs to be focussed on this and it requires more discussion about how users will run the model beyond what is presented in the paper, particularly with the new climate variables that will be included, how a new region is considered, how a new RCP is used and what will happen to the model when developments to the simple climate model and global/regional climate model are made. It may perhaps be better placed in a journal such as 'Advances in Statistical Climatology, Meteorology and Oceanography (ASCMO)'.

We believe that it would be best to defer to the editor regarding whether this paper is best suited for publication in GMD or elsewhere.

I will continue the review based on the method development.

General comment:

As a method I think this is a neat way to produce a large ensemble of climate variables at a regional level that would otherwise be unavailable and attempting to encapsulate the structural uncertainty and weather variability. In some places I find the method hard to follow and the paper needs clarity. In some places I think it would benefit from further equations to back up the text. I would also like to see more discussion on how parametric uncertainty could be included or why it is not deemed necessary. I would also like to see how additional RCPs are included in the method – are they included in the final PDFs or do you envisage separate PDFs for each scenario?

Particular points:

1. Introduction:
There is not enough of a literature review here. In particular, the UK Met Office have developed methods to produce probabilistic climate projections with a similar aim to this Paper.

We have followed the reviewers suggestions and look more widely for existing papers that support, or align with, our paper, and have cited an additional 7 papers that relate to our work.

Please discuss how this work achieves the goal differently. It would be good to see the work of Tebaldi and Knutti referenced here too discussing the difficulties in producing probabilistic information from multiple models.
In particular, the following papers should be referenced and discussed.
- Murphy, J. M. et al. A methodology for probabilistic predictions of regional climate change from perturbed physics ensembles. Phil. Trans. R. Soc. A 365, 1993–2028 (2007).
- Sexton, D., Murphy, J., Collins, M. & Webb, M.Multivariate predictions using imperfect climate models: Part 1 outline of methodology. Clim. Dynam. http://dx.doi.org/10.1007/s00382-011-1208-9 (2011).
- Harris, G. R., et al. "Probabilistic projections for 21st century European climate." Natural Hazards and Earth System Sciences 10.9 (2010): 20092020.

We have cited and discussed the pertinent features of the papers mentioned by the reviewer. However, we decided not to cite the review paper by Tebaldi and Knutti (2007) as their study focuses on multi-model ensembles, which is a different approach to generating probabilistic projections of climate from the methodology presented in our paper. With citing

the other studies, we have included an overview of other methods that have been used in the past to generate climate projection.

Line 2: 'will correctly simulate that trajectory'. I think the inclusion of 'correctly' is necessary since all climate models are attempting to simulate this trajectory.
Corrected.

2.1 Regional Climate Model
How dependent on the RCM are the relationships that are established? Would the relationships be expected to change with a new RCM?
The relationship between the predictors and the predictants is linear, and the regression model fit coefficients should, ideally, be robust properties of the climate system and should not depend on the RCM being used. However, to accommodate the spread of sensitivities of climate variables to changes in global mean temperature due to different model parameterisations, we combine the obtained relationships from different RCM simulations as described in the paper.

Page 3, Line 10: What does 'adequate' mean?
The resolution of the RCM is sufficient to describe the large scale processes New Zealand faces.

Page 3, Line 21: 'The model simulates all atmospheric and land surface processes'. There are missing processes and those that are included are subject to uncertainties – is the method robust to this?
In this study we used simulations from a single RCM which was forced by a number of different AOGCM realizations (different AOGCM boundary conditions for each simulation). Therefore, if the processes are not included in the RCM then these will not impact the structural uncertainties of the ensemble. The missing processes and uncertainties in AOGCM models are what play a role in determining the structural uncertainty in the ensemble.

The effect of the different processes and model parameterisations of RCMs could be assessed by applying this method using a training set which included a number of different RCMs. The methodology would still hold.

Page 3, Line 26: What is the implication of the remaining bias?
The RCM data are used to model how the variable of interest and its variability change over time, while the baseline climatology is obtained from observations. The static and time varying components of the ensemble members are generated separately. Any remaining bias in the RCM data is removed when anomalies are calculated therefore it does not influence the ensemble members.

2.2 Simple climate model
Is MAGICC the only such model? How is it known to simulate annual mean global mean temperature adequately? Are the 19 AOGCMs and 10 carbon cycle models defined by MAGICC or could a user change what is included?

MAGICC is an open access model, publicly available from (incl. website) which has been widely used in the scientific community, but is not the only simple climate model that is able to produce annual mean global mean temperature. The tuning files are provided with MAGICC, but if the user needs additional tunings (other AOGCMs or carbon cycle combinations), additional model tunings can be created.

Page 4, Line 2: It is you that considers the 190 simulations as equally probable. 'We consider the resultant 190 different 'tunings' for MAGICC to be equally probable.' Could a user make different decisions?
Yes, a user could make different decisions if they had additional insight into which simulations could be more probable than others. The method we have developed can easily accommodate this more sophisticated approach. For the purposes of demonstrating our methodology, we have simply followed the methodology of Reisinger, A.; Meinshausen, M.; Manning, M. and Bodeker, G., Uncertainties of global warming metrics: CO2 and CH4, Geophysical Research Letters, 37, L14707, doi:14710.11029/12010GL043803, 2010. We do not have additional insight from the MAGICC team as to which tuning files, both for the AOGCM emulation, and for the carbon cycle emulation, would be more probable than others and so could not encompass such additional information in our demonstration of the methodology. But there is no reason why that would not be possible if that information was available.

Page 4, Line 3: What does 'some' mean? Since the title of the paper states that the structural uncertainty is encapsulated this needs explaining.
The annual mean global mean temperatures from MAGICC do not include inter-annual variability. An additional sentence has been added to the manuscript clarifying this limitation.

Page 4, Line 7: This is the first mention of 'local' -I think this should be included earlier as it's an important point. Earlier, X is used. It would add clarity to define X better earlier on and use X here and in future.
We have made this addition.

2.3 Virtual Climate Station Network
Is this the best observational dataset beyond New Zealand?
No. The VCSN data set is for New Zealand only - we have now clarified that in the paper. People wanting to deploy this method for other countries would need to source their own version of such a data set.

3. Methodology
This section would benefit from some equations to clarify the text – especially right at the beginning to go with the bullet points.
We feel that adding additional equations at this point in the manuscript would likely confuse readers. The following sentence states that more detailed descriptions are given below and the purpose of this section is to provide a high level overview of the methodology.

Why is the period 1960-2100 used?

Once the model has been trained on RCM data, which span 1971-2100, ensemble members can be generated over any reasonable timeframe for which SCM output can be produced. This assumes that the climate system continues to respond linearly, which may not be true if applied under conditions well outside those for which the model was trained on. The period 1960-2100 was chosen arbitrarily to suit our later analysis.

Page 4, Line 27: 'one or more RCM simulations' – I don't quite understand how the RCM simulations are being used here? Later it says you use five – what is the benefit of five and how did you choose these five? How could a user choose a different number?
The methodology is agnostic to the number of RCM simulations available. The 5 simulations that we used were produced using the same model and were all that were available for this study at the time of writing. The uncertainty in the response of the RCM model can be explored even further if more RCM simulations were available.

Page 4, Line 28: You haven't defined alpha yet so this is confusing. Can you state what the sampling of these alpha values is representing instead?
We have added a clarifying sentence.

Page 5, Line 2: 'valid for any GHG emissions scenarios' – is it valid if the regression isn't robust? How can a user validate this if they change the GHG scenario?
No, the method should only be performed if a robust regression fit is obtained. This caveat has been added to the manuscript.

Page 5, Line 2: Why is the anomaly calculated to the 2000-2010 baseline? You later say it is rather short so please say why you have chosen it.
The baseline period was chosen with respect to the application for which we used the data. We were interested in assessing how maximum and minimum surface temperatures change over the 21st century. A sentence describing this choice of the baseline period was added to the manuscript.

3.1 Climatology
Page 5, Line 9: Are you following the method cited or have you expanded it? Can you say why you have chosen this particular method to account for seasonality?
We are following the method presented in Kremser et al. (2014) and expand the fit-coefficients in Fouriers to account for seasonality. Another method would be to fit the regression model completely independently for each month, i.e. first fit the model to only the January data, then fit the model to only the February data etc.. You will then end up with 12 regression coefficients, one for each month, that capture the seasonal dependence of the data on the basis functions. The disadvantage of this approach is that the number of fit coefficients increases by a factor of 12. This significantly increases the uncertainty on the fit coefficients. Using Fourier expansions to account for seasonality reduces the number of fit-coefficients to 5 and therefore reduces the uncertainty.

Figure 2:
I think magenta and red are mixed up.
Corrected.

3.2.1 Training phase
I find this section particularly hard to follow and think it would benefit from more precise equations. Equation 2 doesn't help to explain what you actually did in terms of using time series from 5 RCMs.
We acknowledge the reviewers concern that this section is hard to follow and have reordered the section and provided more background information about the process that is being undertaken. Specifically, we discuss how each RCM simulation is handled independently and have included an additional equation (Eq. 3) with the fit coefficient term expanded.

How have you built a model to explain the relationship between variables of a different scale (daily and annual)? How robust is this relationship with only 5 RCMs? Can you explain the statistical models that you build and how they are validated to produce robust relationships that can be used with the simple climate model output?
A majority of the uncertainty in the ensemble arises from the the range of $T'_{global}$'s produced by MAGICC. The regression from the 5 RCMs agree relatively well for a given GHG emissions scenario. More information regarding the regression model used is provided in Bodeker and Kremser (2015) and Moore et al. (2003). This shortcoming has been addressed in the revised manuscript.

Page 7, Line 6: The relationship itself is calculated over the period 1960-2100?
No, during the training phase, the relationship is calculated using the RCM data sets which span the 1971-2100 period.

Page 7, Line 7: 'an annual mean global mean ..temp.. series' – do these match the period of X'? Are there five series from the five models?
Correct. This is discussed in the paragraphs that follow.

Page 7, Line 7:
\alpha(d) is the fit coefficient – are there 365 of these values?
No there are not 365 alpha values. As explained in the manuscript, there are 5 fit coefficients values, $\alpha_0$ to $\alpha_4$ as given in Eq. 3. We have now provided an additional equation and text to clarify this.

Page 7, Line 13: GCM 'and' RCM.
Done.

Page 7, Line 17: I don't see how the Fourier pairs are embedded in equation 2 to find five fit coefficients from 365?
An additional equation has been added to clarify the expansion into five fit coefficients.

Page 7, Line 18: 'recall, that this is applied at..' – I don't see where it told me that is is applied for each RCM simulation – are they not all used to find the relationship? How do you choose a specific RCM simulation? You should also remind the reader at the start of this section that this is done at a grid box level.

This has been addressed in the revised manuscript by reordering of section 3.2.1.

Figure 3: Could you zoom in to show what the red line looks like? It's quite hard to see. Equation 2 must be more complicated than it looks to produce this line.

The interannual variability arises from the $T_{global}$ time series. The $\alpha$ values do not change from year to year, but simply scale the $T_{global}$ values. A sentence has been added to this effect.

Page 8, Line 5: 'The unitless \alpha^'– a is five values obtained from 365 (maybe more for the five RCMs) so I can't picture what it really is or how it represents sensitivity to temperature. I'm sorry I'm quite lost with respect to alpha. Maybe more equations would help.

For a given RCM, the regression model fit-coefficient $\alpha$ represents the relationship between the variable of interest X' and $T'_{global}$: X'(t) = α x T'global(y)

To capture the seasonal dependence of X' and $T'_{global}$, the fit-coefficient α is expanded in Fourier series (as described in Eq. 1). We have now include another equation into the revised manuscript to clarify this expansion and to clarify the resulting 5 fit-coefficients:

$$X'(t) = (\alpha_0 + \alpha_1 \times \sin(2\pi\, d/365) + \alpha_2 \times \cos(2\pi\, d/365) + \alpha_3 \times \sin(4\pi\, d/365) + \alpha_4 \times \cos(4\pi\, d/365)) \times T'_{global}(y) + R(t)$$

Page 9, Line 6: alpha depends on the RCM. How have you used the RCMs here? Have you chosen one – if so, why start with five? What happens to the data from the other 4? Have you found alpha with all five and then randomly selected from all of them to produce the MC sample? Perhaps reordering the writing here might clear things up with regards to how you are using the RCMs.

The next sentences describe how the alpha value is chosen. We generate 5 alpha values, one for each RCM simulation available. Each ensemble member randomly chooses one of the 5 alpha values.

3.3 Indirect response.
Page 10, Line 26: What does series mean here? Please be more precise.

We have removed "a series of daily maps of" as we agree that it confuses the explanation of EOF analysis. This paragraph has been reworded for clarity.

Page 11, Line 4: I don't understand this point. Can you reword it to be more precise.

We have followed the suggestion by the review and reworded the sentence to:
"That there will be patterns of variability (weather) whose amplitude and variability will respond to climate change as well as others which do not change with increases in $T'_{global}$."

3.3.1 Identifying the modes.
Page 11, Line 6: Are you referring to the residuals in Figure 3? Your language appears to have changed here and it sounds like you are doing something new. If these are the residuals please use the same language and link it better to the previous sections.

Figure 3 shows the anomalies of the daily surface temperature. The residuals are what remains unexplained by the fit of Eq. 2. This sentence has been reworded to expressly mention where these residuals come from.

This sentence has been corrected.

The first EOF means that the mode common mode of variability in New Zealand is that the entire country is warmer or colder than average on a given day. The corresponding principal component time series shows a strong correlation with $T_{global}$ (not shown). EOF 2, 3 and 4 represent different large scale weather patterns typically seen in New Zealand with the Southern Alps (Middle of the South Island) causing differences in East-West and North-South. For example, EOF2 shows that areas of the North Island and East coast of the South Island are often anomalously warm (or cold if the PC is negative) on the same day.

Figure 5 presents an example of the EOF output for one particular RCM simulation which has been forced by the sea surface temperatures from the NorESM1-M mode. The same analysis has been performed for the 4 other RCM simulations using the boundary conditions from other AOGCMs as detailed in the manuscript (not shown). All examples use the simulation that was forced by prescribed RCP 8.5 NorESM1-M sea surface temperatures for consistency.

The period 1972 to 2013 is used for the VCSN data only (because the observations are only available over that period), while for the RCM simulations output till 2100 can be used.

This would hold everywhere, assuming a suitable observational dataset was obtained for the location of interest.

This is the same process as described in the training phase (Sec 3.2.1), but in this case the X' is replaced with PC and α is replaced with β.

$$PC\_pseudo(t) = \beta \times T'_{global}(y) + R(t)$$

By transforming the α basis function using a first order autoregressive model as described in Bodeker and Kremser (2015)

Page 13, Line 25: Why did you not remove autocorrelation at larger time lags? Can you talk about the implications? If it would matter that the interannual variability is too small why not calculate and remove it? If it's not 'too small' then justify not doing it.

We do not have a suitable proxy for modelling how this interannual variability changes with time and annual mean global mean surface temperature.

 How did you create pdfs from time series? Is the data from the whole time series in the pdf or are these time slices? What is the implication of having time series data in these pdfs? Have you got a sample of PCs to create the pdf?

As mentioned in the caption for Figure 6, these PDFs are generated from time slices (2000-2010 and 2090-2100) for each PC time series. Choosing relatively short time slices ensures that any temporal trends in the data do not skew the PDFs.

**3.3.3 Modelling higher order modes.**

 Can you better explain where \sigma comes from? Perhaps use an equation.

This equation has been restructured and references the equation describing fourier pairs in an earlier section

 Why is n=50?

The first 50 EOFs explain approximately 98% of the variability in the weather noise which was deemed sufficient to capture the patterns of variability. The explained variability in each subsequent EOF rapidly decreases.

Figure 7: Can you show a close up of one of the blue lines perhaps – it's difficult to see how the smaller scale temporal patterns are captured and discussed later.

The blue line shows the median of 1900 daily temperature values with small but growing annual cycle over the period of interest. The importance of the blue line is to show the long term trend over the 21st century, rather than showing a close up of the annual cycle. A close up of the blue line does not add significant information to the paper

 I don't understand the line starting 'The apparent annual cycle..'. Could you elaborate? Is this what you'd expect and is it adequate?

To clarify, the annual cycle seen in Figure 7, is not an annual cycle in the temperature anomalies (because the annual cycle has been removed as stated in the figure caption), but rather represents that the variability is changing over time.

Yes, this is an expected result due to plotting daily data as percentiles for a long time span.

 Are there any implications to the interannual variability being smaller? Is improving this a future direction?

The inter-annual variability limits the utility of analysing each ensemble member in isolation. We would like to address this limitation in future research.

 Can you say where 19000 comes from? Previously, you mentioned 1900 simulations so I'm unsure what the extra simulations are taking into account.

The Monte Carlo analysis and modelling of the weather noise are stochastic processes. Each model run produces a different set of ensemble members. Therefore, a large number of ensemble members can be generated drawing from the same statistical relationships

established in the training phase. In this case, 10 ensemble members were generated for each $T_{global}$. Clarifying sentences have been added to the manuscript.

Technical points: Abstract: Page 1, Line 11: change the direction of the first quote mark. This happens in other parts of the paper too.
Corrected.

Introduction: Page 2, Line 7: Do you mean 'uncertain'?
Removed the word.

Page 4, Line 28: Make all of global subscript – use _{global}.
Fixed.

**Anonymous Referee #2**

We would like to take this opportunity to also thank this reviewer for their thorough review of this paper. Their suggested changes have certainly improved the quality of the paper.

Jared and colleagues present a method to compute distributions of local weather variables, and provide an example of how it can be applied to the case of New Zealand.
For each geographic location, the method generates combinations of local climatology, long-term forced changes, and stochastic weather. By combining long-term temperature projections consistent with a large set of AOGCMs and carbon-cycle models, the paper claims to encapsulate model structural uncertainty in ensemble projections of future climate, and climate change.

The method proposed by the paper is interesting. However, the paper neglects references to earlier literature that have proposed related approaches. Furthermore, the method as it is currently described shows some serious shortcomings, particularly in the assessment and inclusion of structural uncertainty in the Probability Density Functions (PDFs) suggested. Two issues stand out:

1) Rather than encapsulating structural model uncertainty in a sensible and robust way, the current method basically multiplies and preserves model sampling bias. Just like the proposed method explores stochastic weather variations with an EOF analysis to understand dominant modes of variability, the same should be carried out for the 19 AOGMs and carbon-cycle models.
While what the review states is, in principle, possible, this is not the purpose of this analysis. Nobody has run all 190 combination of 19 different AOGCM and 10 different carbon cycle models to generate the fields that would be required to conduct such an EOF analysis. Few modelling groups would have the computing power or personnel to achieve that. In an ideal world this would be the optimal way to achieve an ensemble of projections that encapsulates structural model uncertainty in a sensible and robust way - the method we have published provides a practical means of achieving this in a world that is not ideal. It is clear from the

description of our method that it does not solve **every** aspect of the model structural uncertainty problem, but we are not aware of any method that does.

The implicit assumption that each AOGCM realization is statistically or structurally independent is not supported.

Our method does not assume that each AOGCM realization is statistically or structurally independent - noting that we have no AOGCM realizations. It is therefore not immediately obvious to us what the reviewer's criticism is. To be clear: our method relies on the existence of 19 different AOGCM tuning files for the MAGICC simple climate model and 10 different carbon cycle model tuning files for MAGICC. We acknowledge that neither set of tuning files spans the entire potential tuning space and we acknowledge that the set may not reflect the true distribution of the most probably tunings. We make no pretence that we achieve these goals and we are not aware of how would could achieve them; one would need to know, for example, what the exact PDF for climate sensitivity looks like. We have now added the paragraph.

"The EPIC method does not attempt to faithfully represent the full, true PDF of potential tuning parameters both for the AOGCM tunings and the carbon cycle model tunings i.e. were MAGICC tuned to a different set of AOGCMs (e.g. the CMIP5 set rather than the CMIP3 set), we would obtain a different set of tuning files which could lead to a somewhat different spread in our generated ensembles. The purpose of this paper is not to generate perfect ensembles that encapsulate structural model uncertainty in a completely accurate way but rather to describe a method that provides a better representation of that uncertainty than can be achieved with only a limited set of RCM simulations. The robustness of the EPIC method depends on the set of AOGCM and carbon cycle model tunings available and as more comprehensive sets (that better reflect the likelihood of some tunings over others) become available, we expect that the large ensembles generated by EPIC to better reflect the true underlying uncertainties."

This would benefit strongly from appreciating the findings from Masson & Knutti (2011) or Knutti et al (2013) to determine the structural independence of AOGCMs, and apply the methods of model weighting as described in Knutti et al (2017) in order to address structural model uncertainty in a meaningful way.

We have cited and discussed the pertinent features of the papers cited by the reviewer. While we see the value in the weighting of projections in the generation of multi-model ensembles as described in Knutti et al (2017), we feel that this method does not apply. The annual mean global means obtained from MAGICC do not provide enough information to develop a robust model quality metric. Not all combinations of AOGCM and carbon cycle models are available in CMIP3, therefore, a model quality metric cannot be established from AOGCM output for every MAGICC $T'_{global}$.

2) The proposed method hinges on the assumption that fields of variable X are independent of the structural model uncertainty in AOGCMs.

This is not true. The proposed method hinges on the assumption that the model structural uncertainty (both for AOGCM and carbon cycle model) is reflected in the 190 annual mean global mean surface temperature time series that we use as predictors to generate fields of

X. Quite the opposite from what the review states is true, our reconstructions of fields of X are completely dependent on the structural uncertainties of the AOGCMs and carbon cycle models whose combinations of tunings were used to create the 190 member set of $T'_{global}$. Our assumption is that the distribution better reflects that PDF of the resultant ensemble projections of X than would a limited set of output from RCM simulations.

This assumption is not supported by any evidence.
We agree that if we had made this assumption, that it would not be supported by the evidence.

Not all patterns have to be equally probable to occur with a certain T_global response to a specific forcing path.
We agree and have captured this by having the principal component time series correlate with $T'_{global}$ where such a correlation is found to be statistically robust. This allows for the probability of the patterns to change with changing $T'_{global}$ in a way that is consistent with RCM projections.

What is required here is evidence based on a re-analysis of AOGCM data which shows that local patterns (or patterns of boundary conditions for the RCM) are either equally probably across high and low-response AOGCMs, or differ across these responses pointing towards the limitations of the proposed method.
Our analysis does not use AOGCM data. It uses RCM data because we are focussing on an RCM domain. And we do indeed analyse the RCM data for their long term change in the probability of different patterns occurring.

The claims about the applicability and usefulness of the method would be unsupported if both these points are not addressed in a significantly revised manuscript.
We have now added material and clarification to address these points, where appropriate, in the paper.

Smaller editorial comments:

P4L28: T_global is formatted incorrectly
Corrected.

P5L2: Please edit this sentence for spelling and grammar. The authors need to provide evidence to make the claim that the methodology is valid for any chosen GHG emissions scenario.
Corrected.

P6Fig2: Color descriptions do not match the figure.
Corrected.

[revised manuscript text omitted]

---

## Author Response (AR2)

For clarifying our answers to the reviewers' comments, the following scheme is used: Comments of the reviewers are denoted in blue while the authors response is given in black. The changes made to the manuscript in response to the reviewers comments have been appended.

We would like to thank the reviewer for their time and input that they have provided. This has lead to a number of improved the quality of this paper. However, we believe that the reviewer has misunderstood one or two key facets of our methodology. This is no fault of the reviewer's but rather ours in not explaining our procedure clearly enough. Fundamentally, it is not the purpose of this paper to validate the assumptions central to climate pattern-scaling. This has been done in a number of previous publications, viz. Huntingford et al., 2000; Kennett et al., 2006; Kremser et al., 2014; Mitchell et al., 1999; and Mitchell, 2003 to which we refer the reviewer. We have therefore revised the paper to provide greater clarity.

REVIEW (by reviewer #2):

Lewis and colleagues have provided responses to all of the points raised in my first review. However, their responses are not satisfactory. My main concern remains, related to the potential geophysical inconsistency which the authors are introducing between global AOGCM emulations and regional RCM simulations driven by AOGCM boundary conditions.

First we interpret the reviewers concern as a concern as to whether the $T'_{global}$ times series obtained from:

1. These 6 AOGCM CMIP5 models used to provide the SSTs which are later corrected to provide the surface boundary conditions for the HadAM3P model which then provide the lateral boundary conditions for the RCM,
2. The resultant 6 HadAM3P simulations which use corrected SSTs, and
3. The $T'_{global}$ time series obtained from MAGICC simple climate models.

are geophysically consistent. This is indeed a valid concern. First we point out that geophysical consistency between (1) and (2), and between (1) and (3) is not necessary. It is (2) that is used in the training of EPIC and (3) that is used in the implementation of EPIC. The $T'_{global}$ time series between those two data sources do indeed need to be geophysically consistent. To be comprehensive, we have compared the $T'_{global}$ time series obtained from the HadAM3P simulations, with the $T'_{global}$ time series obtained from the CMIP5 AOGCMs that provided the SST boundary conditions for the HadAM3P simulations, as well as the 190 $T'_{global}$ time series obtained from MAGICC. This comparison is shown in Figure 1 below. There are clear differences between the $T'_{global}$ time series obtained from the CMIP5 AOGCMs and those obtained from the HadAM3P simulations. This is because the SSTs from the CMIP5 AOGCMs are bias corrected (downwards) before being used as the surface boundary conditions for the HadAM3P simulations. The 6 $T'_{global}$ time series from the HadAM3P simulations (used in the *training phase* of EPIC) fall well within the envelope of the 190 MAGICC $T'_{global}$ time series used in the *implementation phase* of EPIC, even though MAGICC is emulating a range of CMIP3 models. There is therefore no need for the reviewer to be concerned that there is a geophysical disconnect between the $T'_{global}$ time series used in the training of EPIC and the $T'_{global}$ time series used in the implementation of EPIC.

[Figure]

**Figure 1:** Annual mean global mean surface temperatures calculated from the CMIP5 AOGCM simulations under the RCP8.5 scenario (coloured solid lines). The annual mean global means from the HadAM3P (dashed lines) and MAGICC (grey lines) for RCP8.5 are also shown.

We have created a new version of Figure 1 to make this flow clearer and have added Figure 1 (above) and its associated material to the paper as requested by the reviewer.

The authors mention in their response that they „have no AOGCM realizations".

This is correct. We have only RCM output obtained from HadAM3P simulations, forced by corrected SSTs obtained from simulations of 6 AOGCMs from the CMIP5 archive i.e. BCC-CSM1.1, CESM1-CAM5, GFDL-CM3, GISS-EL-R, HadGEM2-ES and NorESM1-M (selected for their ability to best simulate changes in synoptic scale climate around New Zealand).

My understanding from a careful read of the manuscript is thus that they are using 19 AOGCM tuning files (apparently being used without knowledge of the underlying AOGCM data)

The 19 AOGCM tuning files were derived by the MAGICC team by minimizing differences between global and hemispheric (differentiated into land and sea) MAGICC diagnostic variables and the same variables derived from the AOGCM simulations. Therefore these tuning files are being used in a way that is entirely consistent with the underlying AOGCM data.

together with AOGCM boundary conditions for their RCM derived from one AOGCM (HadAM3p). Their response suggests that they consider using the global mean response from 19 different AOGCMs (represented by their tuned MAGICC emulations) to scale the regional response is a physically acceptable approach.

That's correct. This is the very basis of climate pattern-scaling.

In my earlier review I requested evidence for the robustness of this assumption.

The first step towards demonstrating this robustness has been provided above in Figure 1. Further we note: in the training phase we derive $\alpha$ coefficients by fitting:

$$T(i,j) = \alpha(i,j) \times T'_{global} + Weather$$

to temperatures from RCM output (T(i,j)) using $T'_{global}$ time series from the HadAM3P simulations. It is entirely correct to use the $T'_{global}$ time series from the HadAM3P simulations for this purpose since it was those simulations that were used to provide the lateral boundary conditions for the nested RCM simulations. Then, in the implementation phase we use $T'_{global}$ time series from MAGICC (shown in Figure 1 above to be geophysically consistent with the $T'_{global}$ time series from the HadAM3P simulations) to reconstruct T(i,j) from the $\alpha(i,j)$ coefficients we have derived. The only assumptions we have made are:

1. The $T'_{global}$ time series used in the training of EPIC are geophysically consistent with the $T'_{global}$ time series used in the implementation of EPIC.
2. The $\alpha(i,j)$ coefficients derive their value from the slow secular correlation between T(i,j) and $T'_{global}$ and not from any correlation accruing from year-to-year correlation between these two quantities. This is important because the MAGICC simulations of $T'_{global}$ are smooth and contain no year-to-year variability.

Evidence for the robustness of assumption (1) is provided through Figure 1 above. To validate assumption no. (2), we have calculated $\alpha(i,j)$ both from smoothed and unsmoothed $T'_{global}$ time series obtained from the HadAM3P simulations. The smoothing was performed using a Savitzky-Golay filter with a window length of 15 years and a polynomial order of 3. The resultant changes in $\alpha(i,j)$ are insignificant (see Figure 2).

[Figure]

**Figure 2:** The annual cycle of the $\alpha$ regression coefficient obtained from Eq. 2 in the revised paper for daily maximum surface temperature in Auckland, New Zealand using an unsmoothed (blue) and smoothed (green) $T'_{global}$. The smoothing was performed using a Savitzky-Golay filter with a window length of 15 years and a polynomial order of 3. The data used in the training of this regression model were obtained from AGCM and RCM simulations which were derived from a CMIP5 BCC-CSM1.1 AOGCM simulation under RCP8.5.

The authors indicate that running a full 190 member ensemble with 19 AOGCMs and 10 carbon-cycle models is computationally impossible.

We would like to point out that even if it were possible, there would be no reason to do it because our method in no way requires it.

I agree. However, there is a much simpler method which allows to illustrate the internal consistency between the AOGCM response assumed in MAGICC

We assume here that by 'response' the review means the $T'_{global}$ time series generated by MAGICC for some emulated AOGCM under some GHG emissions scenario.

and the boundary conditions used for their RCM, based entirely on data already available in the public domain. The internal and physical consistency of joining global AOGCM emulations with regional patterns of a single RCM-AOGCM pair can be shown by simply computing the boundary conditions for the RCM domain that would be implied by each of the CMIP3 AOGCMs that is emulated.

But that would show us nothing because we would not use the CMIP3 AOGCMs to provide the boundary conditions for the RCM domain. The SSTs would likely be wrong. Therefore we train EPIC on $T'_{global}$ time series obtained from HadAM3P simulations which use corrected SSTs and then, in the implementation phase of EPIC use $T'_{global}$ time series obtained from MAGICC which have been shown (see Figure 1) to be internally consistent with the HadAM3P simulations. Previous work we have performed has demonstrated that the derived $\alpha$(i,j) coefficients are insensitive to the GHG emission scenario use for their derivation (Kremser et al., 2014). We also point out that we derive our $\alpha$ coefficients not just from a single RCM-AOGCM pair, but from 6 simulations of the HadAM3P model with the nested RCM.

Data for this is available at the CMIP repository. This exercise can be carried out for a small set of salient parameters, either at a given warming level, or at a given concentration level/time slice (for example if a 2xCO2 experiment is used). A statistical analysis of these 19 possible RCM boundary conditions and comparison of these statistics with the actual boundary conditions used for the RCM will show whether there are significant differences between them.

There will most definitely be differences between them since the boundary conditions for our RCM simulations came from HadAM3P simulations which used corrected SSTs from the CMIP5 AOGCM simulations. So we would not be able to infer anything from this comparison.

"Significant" can here either mean statistical significance, but it can also involve expert judgment about how much these variations would actually matter.

If there are no significant differences between the fields, the authors have provided the necessary evidence that data is combined in a physically plausible way.

We disagree. By not using $T'_{global}$ time series from CMIP5 AOGCM simulations in the training phase of EPIC, and rather using the $T'_{global}$ time series from a single HadAM3P model (6 different simulations) and then using that same single model to provide the lateral boundary conditions for our nested RCM, we believe that we are combining the data in a physically plausible way. In addition, we have verified that the $T'_{global}$ time series used in the training phase are physically consistent with the $T'_{global}$ time series used in the implementation phase. We cannot see that any further justification of the method is required.

However, if there are significant statistical differences between the RCM boundary conditions derived from the 19 CMIP3 RCMs and the HadAM3p boundary conditions, the authors either need to clearly argue that these differences would not affect the EOFs used at the regional scale

Such a difference would have no effect whatsoever on our derived EOFs because these come from the RCM driven by HadAM3P boundary conditions not from CMIP5 AOGCM boundary conditions and it is the $T'_{global}$ time series from those HadAM3P simulations that are then used to reconstruct weather variability from the EOFs. Again, this is all internally geophysically consistent - see agreement of first 4 series of EOFs in Figure 6. Furthermore, we correct for the fact that the RCMs are unlikely to generate weather correctly (see Figure 7 in the revised paper).

by referring to the earlier literature, or accept that their method implies an unphysical combination of global-mean AOGCM responses emulated by MAGICC and regional responses simulated by their RCM. In the latter case, this caveat should be made appropriately and its implications for the applicability and robustness of the method should be discussed up front.

We have revised the paper in the hope that the EPIC methodology is now more transparent.

Huntingford, C. and Cox, P.M.: An analogue model to derive additional climate change scenarios from existing GCM simulations, Clim. Dynam., 16, 575–586, 2000.

Kennett, E.J. and Buonomo. 2006. Methodologies of pattern scaling across the full range of RT2A GCM ensemble members, report from ENSEMBLE-based Predictions of Climate Changes and their Impacts.

Kremser, S., Bodeker, G.E., and Lewis, J.: Methodological aspects of a pattern-scaling approach to produce global fields of monthly means of daily maximum and minimum temperature, Geosci. Model Dev., 7, 249-266, 2014

[revised manuscript text omitted]